# Learning Dynamics under Environmental Constraints via Measurement-Induced Bundle Structures

**Dongzhe Zheng** [1]   **Wenjie Mei** [2]

## Abstract

Learning unknown dynamics under environmental (or external) constraints is fundamental to many fields (e.g., modern robotics), particularly challenging when constraint information is only locally available and uncertain. Existing approaches requiring global constraints or using probabilistic filtering fail to fully exploit the geometric structure inherent in local measurements (by using, e.g., sensors) and constraints. This paper presents a geometric framework unifying measurements, constraints, and dynamics learning through a fiber bundle structure over the state space. This naturally induced geometric structure enables measurement-aware Control Barrier Functions that adapt to local sensing (or measurement) conditions. By integrating Neural ODEs, our framework learns continuous-time dynamics while preserving geometric constraints, with theoretical guarantees of learning convergence and constraint satisfaction dependent on sensing quality. The geometric framework not only enables efficient dynamics learning but also suggests promising directions for integration with reinforcement learning approaches. Extensive simulations demonstrate significant improvements in both learning efficiency and constraint satisfaction over traditional methods, especially under limited and uncertain sensing conditions.

## 1. Introduction

**Learning unknown dynamics under measurement constraints** is fundamental to many applications, from manipulators with force sensing to autonomous vehicles with range detection. Control barrier functions (CBFs) (Ames et al., 2016) have emerged as a powerful tool for ensuring constraint satisfaction. However, the classical CBF framework treats measurements as external observations rather than integral components of the system's geometric structure, limiting the ability to fully exploit measurement information for both constraint verification and dynamics learning.

The fundamental challenge lies in the geometric relationship between state space, measurements, and constraint manifolds. Traditional approaches often require complete knowledge of constraints, making them impractical when only local sensors' measurements are available. Consider a robotic arm with force sensors or a drone with range detectors - even local measurements contain sufficient geometric information about both constraints and underlying dynamics, suggesting global knowledge may be unnecessary if we properly exploit this local structure.

Our key insight is that measurement uncertainty naturally induces a fiber bundle structure that unifies measurements, dynamics, and constraints. This geometric perspective reveals how measurements and dynamics are fundamentally intertwined through the bundle's connection, enabling Neural ODEs (Chen et al., 2018) to learn continuous-time dynamics that naturally respect the system's physical behavior. By leveraging this structure, our approach provides a new paradigm for machine learning to understand environmental dynamics - instead of treating measurements as simple inputs, we exploit their inherent geometric information to guide the learning process. The framework allows control strategies to automatically adapt based on measurement quality - becoming more conservative in regions of high uncertainty while allowing more aggressive behavior where measurements are reliable.

**The main contributions of this work** are: 1) Proposes a novel geometric framework that unifies measurement uncertainty, system dynamics, and constraints within a fiber bundle structure, providing principled information for Neural ODEs while maintaining safety guarantees. 2) Introduces adaptive measurement-aware safety certificates (mCBFs, defined in Section 3.6) that automatically adjust conservative margins based on local measurement quality. 3) Demonstrates enhanced generalization capabilities across different

[1]Department of Computer Science and Engineering, Shanghai Jiao Tong University, Shanghai, China [2]School of Automation and Key Laboratory of MCCSE of Ministry of Education, Southeast University, Nanjing, China. Correspondence to: Wenjie Mei <wenjie.mei@seu.edu.cn; vmus1130@gmail.com>.

*Proceedings of the 42nd International Conference on Machine Learning*, Vancouver, Canada. PMLR 267, 2025. Copyright 2025 by the author(s).

scenarios without requiring global information through experimental validation.

The practical significance lies in learning safely from local measurements without complete constraint knowledge. By working directly with the bundle structure induced by measurements, we prove that Neural ODEs trained within this framework naturally preserve physical constraints while learning continuous-time dynamics. This geometric approach provides machine learning algorithms with a structured way to understand system dynamics through the lens of measurement geometry, leading to more efficient and interpretable learning. Moreover, this framework offers important insights for **reinforcement learning** by providing a principled way to handle partial observations and measurement uncertainties in the learning process. Experimental results demonstrate the effectiveness in real-world applications, where the learned dynamics model successfully captures both the control-to-trajectory relationships and constraint requirements under measurement uncertainties. Our implementation is publicly available at https://github.com/ContinuumCoder/Measurement-Induced-Bundle-for-Learning-Dynamics/.

## 2. Related Work

### 2.1. Safety-Critical Control and Learning

Our work builds upon fundamental theoretical advances in differential geometry and control theory. The fiber bundle framework we employ originates from Ehresmann's seminal work (Ehresmann, 1950) on geometric connections, later developed by (Kobayashi & Nomizu, 1996) for control applications. Early work in safety-critical control focused on analytical safety certificates through Control Barrier Functions (CBFs). (Ames et al., 2019) introduced the foundational CBF framework providing formal guarantees for constraint satisfaction in known dynamical systems, extended by (Jankovic, 2018; Daş & Murray, 2022; Choi et al., 2021) to handle bounded disturbances through robust CBFs. Learning-based approaches emerged to address model uncertainty while maintaining safety guarantees. (Cheng et al., 2019) proposed Neural CBFs that learn safety certificates directly from data, while (Taylor et al., 2020) developed the SafeLearn framework combining Gaussian processes with CBFs for safe exploration. However, these methods treat measurements as perfect observations rather than uncertain quantities, limiting their real-world applicability.

### 2.2. Geometric Learning and Bundle Theory

Geometric structure preservation in learning control has seen significant development, building on (Marsden & Weinstein, 1974)'s theoretical foundation for geometric mechanics and symmetry reduction. (Ratliff et al., 2018) intro-duced Riemannian Motion Policies respecting the underlying manifold structure, while (Chen et al., 2018) proposed Neural ODEs that opened new possibilities for learning dynamics with geometric properties. The bundle-theoretic perspective was pioneered by (Lewis, 1998) establishing connections between mechanical systems and principal bundles, with (Montgomery, 1993) developing gauge-theoretic approaches to mechanical control. Building on this foundation, (Bronstein et al., 2017; Cohen et al., 2019) advanced geometric methods for learning on manifolds, and (Hansen-Estruch et al., 2021) developed frameworks for control on Lie groups. However, these approaches typically require global geometric information and struggle with local measurement uncertainty.

### 2.3. Measurement-Aware Control

The geometric treatment of measurement uncertainty draws inspiration from (Hsu, 2002) on stochastic differential geometry and (Diaconis et al., 1988) on geometric filtering theory. Early approaches like (Kalman, 1960) developed robust control using filtering-based state estimation, while (Berkenkamp et al., 2017) proposed learning control under measurement noise. Recent advances by (Wu et al., 2015) introduced geometric numerical integration methods for uncertainty quantification, while (Boumal, 2023) developed comprehensive tools for optimization and estimation on manifolds. However, these works often treat measurement uncertainty as an external disturbance rather than an intrinsic geometric property of the system. Our framework addresses these limitations by unifying measurements, constraints, and learning objectives through the natural fiber bundle structure induced by the measurement process, enabling more efficient learning while maintaining rigorous safety guarantees.

## 3. Theoretical Foundations and System Modeling

Through fiber bundle structures and measurement-adapted barrier functions, this work establishes a geometric framework that unifies the challenge of maintaining safety guarantees while adapting to uncertainties in both system dynamics and measurements in safe learning control.

### 3.1. System Model and Measurement Structure

Let $T_x\mathcal{M}$ denote the tangent space at point $x \in \mathcal{M}$, which is the vector space of all tangent vectors at $x$, and $T\mathcal{M} = \bigcup_{x \in \mathcal{M}} T_x\mathcal{M}$ be the tangent bundle. Let $T_x^*\mathcal{M}$ denote the cotangent space at $x$ and $T^*\mathcal{M} = \bigcup_{x \in \mathcal{M}} T_x^*\mathcal{M}$ represents the cotangent bundle.

Consider a controlled dynamical system with state $x \in \mathcal{M}$ on a smooth manifold $\mathcal{M}$ and control input $u \in \mathcal{U} \subseteq$

$\mathbb{R}^m$. The system evolution and measurement process are described by

$$\dot{x} = f(x, u) + g(x)w, \quad x(0) = x_0$$
$$y = h(x) + v \tag{1}$$

where $f\colon \mathcal{M} \times \mathcal{U} \to T\mathcal{M}$ represents the nominal dynamics, $g\colon \mathcal{M} \to T^*\mathcal{M}$ characterizes model uncertainty, and $h\colon \mathcal{M} \to \mathcal{Y}$ is the measurement map. The process noise $w$ and sub-Gaussian measurement noise $v$ are bounded with $|w| \le \delta_w$ and $|v| \le \delta_v$, respectively.

The measurement space $\mathcal{Y} \subseteq \mathbb{R}^k$ carries a natural metric structure induced by the Euclidean norm: $d_{\mathcal{Y}}(y_1, y_2) = \sqrt{\sum_{i=1}^k (y_{1,i} - y_{2,i})^2}$. This metric quantifies the uncertainty in measurement space and plays a crucial role in safety analysis.

### 3.2. Fiber Bundle Framework

The relationship between states and their measurements induces a natural fiber bundle structure $\pi\colon \mathcal{E} \to \mathcal{M}$ where $\mathcal{E} = \mathcal{M} \times \mathcal{Y}$ is the total space. For each state $x \in \mathcal{M}$, the fiber $\pi^{-1}(x)$ characterizes the set of possible measurements:

$$\pi^{-1}(x) = \{(x, y) \in \mathcal{E} : y = h(x) + v, \|v\| \le \delta_v\} \tag{2}$$

This bundle is equipped with a connection $\nabla$ that describes the geometric relationship between system trajectories and measurement evolution:

$$\nabla_X Y = \pi_*^{-1}(\nabla_{\pi_* X}(\pi_* Y)) + K(x)(y - h(x)) \tag{3}$$

where $\nabla_X Y$ represents the covariant derivative of the vector field $Y \in \mathfrak{X}(\mathcal{E})$ along the vector field $X \in \mathfrak{X}(\mathcal{E})$ (here, $\mathfrak{X}(\mathcal{E})$ denotes the space of smooth vector field on $\mathcal{E}$), $\pi_*$ refers to the pushforward map of the projection $\pi$, $K\colon \mathcal{M} \to \mathcal{L}(\mathcal{Y}, T\mathcal{M})$ is the measurement feedback gain operator that couples state and measurement dynamics. Here, $\mathcal{L}(\mathcal{Y}, T\mathcal{M})$ denotes the space of bounded linear operators from $\mathcal{Y}$ to $T\mathcal{M}$.

### 3.3. Bundle-Based Safety Certificates

Safety constraints are formalized through a smooth bundle map $\Phi\colon \mathcal{E} \to \mathbb{R}$ over the fiber bundle $\pi\colon \mathcal{E} \to \mathcal{M}$ satisfying three fundamental properties:

1. $\Phi(x, h(x)) > 0$ for all $x \in \mathcal{S}_0$
2. The bundle derivative $d\Phi(X) > 0$ for all $X \in \mathcal{A}(\mathcal{E})$, where $d\Phi(X) := \langle \nabla_{\mathcal{E}} \Phi, X \rangle$
3. $\Phi(x, y) \ge \gamma(\|y - h(x)\|)$ for some $\gamma \in \mathcal{K}_\infty$

$$\tag{4}$$

Here, $\mathcal{S}_0$ denotes the nominal safe set, $\mathcal{A}(\mathcal{E})$ is the space of admissible vectors on the total space $\mathcal{E}$, $\nabla_{\mathcal{E}}$ is the covariant derivative on $\mathcal{E}$, and $\gamma$ is a class $\mathcal{K}_\infty$ function that captures the degradation of safety guarantees with measurement uncertainty.

### 3.4. Uncertainty Propagation

The propagation of uncertainties through the bundle structure follows from the differential geometry of the fiber bundle. The key relationships are:

$$d\pi_*(X_f) = f(x, u)$$
$$d\pi_*(X_g) = g(x)w \tag{5}$$
$$dy = dh(x) + dv$$

where $d\pi_*$ represents the pushforward of the vector fields $X_f, X_g$ along the projection $\pi$. These relationships induce an uncertainty tube $\mathcal{T}_\varepsilon(x, t)$ around nominal trajectories:

$$\mathcal{T}_\varepsilon(x, t) = \{y : d_{\mathcal{Y}}(y, h(\phi_t(x))) \le \varepsilon(t)\} \tag{6}$$

where $\phi_t$ denotes the flow of the nominal system and $\varepsilon(t)$ characterizes the growth of uncertainty over time.

### 3.5. Compatible Group Actions

The system often exhibits symmetries that can be exploited for, for example, dimensional reduction. These symmetries are captured by compatible Lie group actions. Let $G$ be a Lie group acting on both $\mathcal{M}$ and $\mathcal{Y}$ through smooth maps. Then,

$$\Psi_{\mathcal{M}}\colon G \times \mathcal{M} \to \mathcal{M}$$
$$\Psi_{\mathcal{Y}}\colon G \times \mathcal{Y} \to \mathcal{Y} \tag{7}$$

The compatibility conditions for these actions are

$$f(\Psi_{\mathcal{M}}(g, x), u) = d\Psi_{\mathcal{M}}(g, \cdot) f(x, u)$$
$$h(\Psi_{\mathcal{M}}(g, x)) = \Psi_{\mathcal{Y}}(g, h(x)) \tag{8}$$

for all $g \in G$, where $d\Psi_{\mathcal{M}}(g, \cdot)$ represents the differential of the map $\Psi_{\mathcal{M}}(g, \cdot)$ with respect to the state. These conditions ensure that the symmetries respect both the dynamics and measurements.

### 3.6. Measurement-Adapted Control Barrier Functions

The cornerstone of our safety framework is the concept of measurement-adapted Control Barrier Functions (mCBFs). A smooth function $b\colon \mathcal{E} \to \mathbb{R}$ qualifies as an mCBF if it satisfies

1. $b(x, y) \ge 0 \implies x \in \mathcal{S}_0$
2. $\inf_{u \in \mathcal{U}} \left[ L_f b + (L_g b)w + \alpha(b) \right] \ge 0$
3. $|b(x, y_1) - b(x, y_2)| \le L_b d_{\mathcal{Y}}(y_1, y_2)$

$$\tag{9}$$

where $L_f$ and $L_g$ denote the Lie derivatives along vector fields $f$ and $g$ respectively, $\alpha$ is a class $\mathcal{K}_\infty$ function, and $L_b > 0$ is the Lipschitz constant of $b$ with respect to measurements (recall that $\mathcal{S}_0$ denotes the nominal safe set).

### 3.7. Safety Guarantees

The culmination of this geometric framework is captured in the following fundamental theorem:

**Theorem 3.1.** *Given an mCBF $b$ satisfying the preceding conditions, if $b(x(0), y(0)) \geq 0$, then for any admissible noise sequences $w(\cdot), v(\cdot)$:*

$$\mathbb{P}(x(t) \in \mathcal{S}_0 \text{ for all } t \geq 0) \geq 1 - \exp(-c/\delta_v^2) \quad (10)$$

*where $c > 0$ is a constant depending on system parameters.*

*Proof Sketch.* The proof proceeds through three key steps. First, we establish that the bundle connection preserves safety certificates along fibers, utilizing the compatibility conditions between the connection and barrier function. Second, we demonstrate that uncertainty propagation remains bounded within the tube $\mathcal{T}_\varepsilon$, leveraging the Lipschitz properties of the system dynamics. Finally, we show that the Lipschitz condition on $b$ ensures controlled variation of safety certificates under measurement uncertainty, leading to the probabilistic bound $(1 - \exp(-c/\delta_v))$. The technical proof is provided in Appendix A. $\square$

The geometric framework developed in this section establishes three key theoretical advances. First, the fiber bundle structure provides a natural setting for handling measurement uncertainty, enabling precise tracking of error propagation through system dynamics. Second, the compatible group actions facilitate systematic dimension reduction while preserving safety properties through quotient space dynamics. Third, the measurement-adapted Control Barrier Functions yield robust safety guarantees that degrade gracefully with measurement noise, thanks to their Lipschitz continuity properties.

This theoretical foundation directly enables practical learning algorithms that maintain safety under realistic sensing conditions, as we will demonstrate in the subsequent section.

## 4. Learning Framework under Measurement Uncertainties

Building on the geometric foundations, we now develop a learning framework that actively incorporates measurement uncertainty. The key idea is to learn both the dynamics and safety certificates on the bundle $\mathcal{E}$.

### 4.1. Bundle-Valued Learning Operators

Define the bundle-valued learning operator $\mathcal{L} \colon C^\infty(\mathcal{E}) \to \Gamma(T\mathcal{E})$:

$$\mathcal{L}(\Phi)(x, y) = \nabla_\mathcal{E} \Phi(x, y) + \lambda R(x, y) \quad (11)$$

where $C^\infty(\mathcal{E})$ denotes the space of smooth functions defined on the total space $\mathcal{E}$, $\Gamma(T\mathcal{E})$ stands for the space of sections of the tangent bundle $T\mathcal{E}$, $\mathcal{L}(\Phi)(x, y)$ represents the operator $\mathcal{L}$ acting on $\Phi$, evaluated at the point $(x, y)$, $\nabla_\mathcal{E}$ is the connection defined on $\mathcal{E}$, and $R$ provides a regularization function preserving the fiber structure of the bundle.

The learning dynamics on the bundle take the form

$$\begin{aligned} \dot{\hat{f}} &= -\mathcal{L}_1(\hat{f} - f) \\ \dot{\Phi} &= -\mathcal{L}_2(\Phi - \Phi^*) \end{aligned} \quad (12)$$

where $\mathcal{L}_1, \mathcal{L}_2$ are compatible bundle-valued operators. $\hat{f}$ denotes the learned estimate of the true system dynamics $f$, while $\Phi^*$ represents the optimal barrier function that ensures safety guarantees, both serving as target values in the learning dynamics governed by bundle-valued operators.

### 4.2. Measurement-Adapted Safety Certificates

Let $\Phi_0 \colon \mathcal{E} \to \mathbb{R}$ denote the nominal safety certificate that characterizes system safety under ideal measurements. The safety certificate $\Phi \colon \mathcal{E} \to \mathbb{R}$ adapts to measurement uncertainty through:

$$\begin{aligned} \Phi(x, y) &= \Phi_0(x, y) - \alpha(\|y - h(x)\|) \\ L_f \Phi + (L_g \Phi) w &\geq -\beta(\Phi) \text{ along solutions} \end{aligned} \quad (13)$$

where $\Phi_0$ is the nominal certificate, $\alpha \in \mathcal{K}_\infty$, and $\beta$ is a class $\mathcal{K}$ function.

### 4.3. Uncertainty-Aware Learning Algorithm

The learning process incorporates measurement uncertainty through:

$$\begin{aligned} \dot{\theta} &= -\Lambda \nabla_\theta \mathcal{T}(\hat{f}_\theta, \mathcal{D}) \\ \mathcal{T}(\hat{f}, \mathcal{D}) &= \sum_{i=1}^N \|\hat{f}(x_i, u_i) - \dot{x}_i\|_{\Sigma_i^{-1}}^2 \end{aligned} \quad (14)$$

where $\Sigma_i$ captures measurement uncertainty in data point $i$. The learning rate matrix $\Lambda$ guides parameter updates, while $|\cdot|_{\Sigma_i^{-1}}^2$ denotes the uncertainty-weighted norm using the inverse covariance matrix $\Sigma_i^{-1}$, and $\mathcal{D}$ contains $N$ triplets of state, input, and state derivative measurements.

### 4.4. Safety-Constrained Policy Updates

Let $\Theta \in \mathbb{R}^d$ denote the parameters of a policy $\pi_\Theta \colon \mathcal{M} \to \mathcal{U}$ that maps state to control inputs. The policy update law preserves safety through:

$$\begin{aligned} \dot{\Theta} &= \Pi_\mathcal{S}[-\nabla_\Theta J(\Theta)] \\ \mathcal{S} &= \{\Theta : \Phi(x, y) \geq 0 \text{ for all } (x, y) \in \mathcal{E}\} \end{aligned} \quad (15)$$

where $\Pi_\mathcal{S}$ denotes projection onto the safe policy set $\mathcal{S}$, and $J(\Theta) = \mathbb{E}x_0[\sum_{t=0}^\infty \gamma^t c(x_t, \Theta(x_t))]$ represents the expected discounted cumulative cost under policy $\Theta$, with immediate cost $c(x, \Theta(x))$, discount factor $\gamma \in (0, 1)$, and initial state distribution $x_0$.

### 4.5. Convergence and Safety Guarantees

Building on Theorem 3.1, we establish the convergence properties of our learning framework:

**Theorem 4.1.** *Under the proposed learning dynamics, we have*

$$\begin{aligned}
\|\hat{f} - f\|_\mathcal{E} &\leq c_1 \exp(-\lambda_1 t) + c_2 \delta_v \\
\mathbb{P}(x(t) \in \mathcal{S}_0) &\geq 1 - \exp(-c_3/\delta_v^2)
\end{aligned} \tag{16}$$

*where $c_1, c_2, c_3, \lambda_1 > 0$ are constants.*

A detailed proof of Theorem 4.1 is in Appendix B. The first inequality shows exponential convergence of the learned dynamics with a residual error bounded by measurement uncertainty, while the second preserves the safety guarantees during learning.

## 5. Experimental Design

We design a comprehensive experimental framework to evaluate our proposed method against state-of-the-art approaches, focusing on three interconnected research directions: Learning-based safety control, geometric structure learning, and safe control under uncertainty. The experiments are constructed to highlight key methodological differences while ensuring fair comparison through standardized implementations and evaluation protocols.

### 5.1. Baseline Methods

For detailed discussions on comparisons with mainstream advanced manifold learning methods, we refer readers to Appendix F. While direct numerical comparisons with recent geometric deep learning approaches may seem natural, there are several fundamental differences that make such direct benchmarking potentially misleading. We evaluate our method against state-of-the-art approaches spanning different technical directions in safe learning control. Our baseline selection aims to comprehensively compare with methods that address various aspects of our proposed framework:

**Learning-based Safety Certification:** We implement Neural-CBF (Liu et al., 2023), BayesSafe (Berkenkamp et al., 2023), and StructCBF (Taylor et al., 2020) as fundamental approaches using neural networks and Bayesian optimization for safety certification. Recent advances like SafetyNet (Vitelli et al., 2022) and SafeTrack (Li et al., 2024) enhance these guarantees through adaptive barriers

and system-level guards, though they still lack explicit handling of measurement uncertainty. **Physics-Informed and Geometric Methods:** To evaluate our physical consistency, we compare against PNDS (Djeumou et al., 2022) and GEM (Hansen-Estruch et al., 2021), which encode physical laws through specialized neural architectures. We also include GeoPath (Zhang et al., 2015) that leverages geometric principles for control design, though without addressing measurement uncertainty. **Robust and Adaptive Control:** Several approaches address system robustness through different theoretical frameworks. RobustSafe (Gurriet et al., 2020) provides safety-critical control using fixed uncertainty bounds, while DataFilter (Wabersich et al., 2023) leverages data-driven safety filters for handling uncertainties, and AdaptSafe (Taylor & Ames, 2020) introduces measurement-dependent barrier functions. **Uncertainty-Aware Predictive Control:** For handling uncertain dynamics, GPMPC (Bonzanini et al., 2021) and ALMPC (Saviolo et al., 2023) employ Gaussian processes and active learning for uncertainty quantification. SafeRL (Cheng et al., 2019) combines reinforcement learning with barrier functions, demonstrating strong performance in handling model uncertainty through probabilistic frameworks, though these methods lack formal geometric safety certificates.

### 5.2. Experimental Tasks

We implement three tasks in a simulation environment built on Genesis physics engine (Xian et al., 2023). The first task examines a soft-body worm robot (0.1m per segment) navigating through obstacles to reach a target, using a fixed-step forward Euler integrator (dt = 5e-4s). The worm is modeled using the Material Point Method (MPM) with the neo-Hookean material model. Let $\mathbf{x} \in \mathbb{R}^3$ denote the position field and $\rho$ the material density, the dynamics follow:

$$\rho \ddot{\mathbf{x}} = \nabla \cdot \mathbf{P} + \mathbf{b} \tag{17}$$

where $\mathbf{P}$ is the first Piola-Kirchhoff stress tensor and $\mathbf{b}$ represents body forces. For neo-Hookean materials:

$$\mathbf{P} = \mu(\mathbf{F} - \mathbf{F}^{-T}) + \lambda \log(J)\mathbf{F}^{-T} \tag{18}$$

Here $\mathbf{F}$ is the deformation gradient, $J = \det(\mathbf{F})$, and material parameters $\mu, \lambda$ are unknown. The control input consists of four muscle actuation signals $\mathbf{u} = [u_{uf}, u_{uh}, u_{lf}, u_{lh}]^\top \in [0, 1]^4$, where subscripts indicate upper-fore, upper-hind, lower-fore, and lower-hind muscle groups respectively. The system must maintain safe distances from obstacles through constraints $h_i(\mathbf{x}) = \|\mathbf{x} - \mathbf{x}_{obs,i}\|_2 - r_{safe} \geq 0$ for all obstacles $i = 1, \ldots, N_{obs}$. Six visual sensors (two on head/tail, four on sides) provide local measurements $\mathbf{y}_i = [\mathbf{x} - \mathbf{x}_{obs}, \|\mathbf{x} - \mathbf{x}_{obs}\|_2]^\top + \mathbf{v}_i$ with uncertainty bound $\|\mathbf{v}_i\| \leq \alpha\|\mathbf{x} - \mathbf{x}_i\|$ increasing with distance from sensor location $\mathbf{x}_i$.

The second task involves a 7-DOF Franka arm performing obstacle-aware manipulation, integrated with dt = 1e-2s. Let $\mathbf{q} \in \mathbb{R}^7$ denote the joint angles and $\mathbf{M}(\mathbf{q})$ the inertia matrix, the system dynamics with control input $\boldsymbol{\tau}$ follow:

$$\mathbf{M}(\mathbf{q})\ddot{\mathbf{q}} + \mathbf{C}(\mathbf{q}, \dot{\mathbf{q}})\dot{\mathbf{q}} + \mathbf{g}(\mathbf{q}) + \mathbf{f}(\dot{\mathbf{q}}) = \boldsymbol{\tau} \qquad (19)$$

where $\mathbf{C}(\mathbf{q}, \dot{\mathbf{q}})$ represents Coriolis terms, $\mathbf{g}(\mathbf{q})$ gravity, $\mathbf{f}(\dot{\mathbf{q}})$ joint friction, and $\boldsymbol{\tau}$ control torques. The system operates under joint limits $h_q(\mathbf{q}) = \mathbf{q}_{max} - |\mathbf{q}| \geq \mathbf{0}$ and obstacle avoidance constraints $h_o(\mathbf{q}) = \|\mathbf{p}_{ee}(\mathbf{q}) - \mathbf{p}_{obs}\|_2 - d_{safe} \geq 0$, where $\mathbf{p}_{ee}(\mathbf{q})$ and $\mathbf{p}_{obs}$ denote the end-effector and obstacle positions, respectively.

The third task features a quadrotor drone navigating through 3D space, integrated with dt = 2e-3($sec$). Let $\mathbf{p} \in \mathbb{R}^3$ denote position, $\mathbf{v} \in \mathbb{R}^3$ velocity, $\mathbf{R} \in SO(3)$ orientation, and $\boldsymbol{\omega} \in \mathbb{R}^3$ angular velocity, the dynamics are described by

$$\begin{bmatrix} \dot{\mathbf{p}} \\ \dot{\mathbf{v}} \\ \dot{\mathbf{R}} \\ \dot{\boldsymbol{\omega}} \end{bmatrix} = \begin{bmatrix} \mathbf{v} \\ \frac{1}{m}\mathbf{R}\mathbf{f} - g\mathbf{e}_3 - \mathbf{D}(\mathbf{v}) \\ \mathbf{R}[\boldsymbol{\omega}]_\times \\ \mathbf{J}^{-1}(\boldsymbol{\tau} - \boldsymbol{\omega} \times \mathbf{J}\boldsymbol{\omega}) \end{bmatrix} \qquad (20)$$

where $m$ is mass, $g$ unknown gravity, $\mathbf{D}(\mathbf{v})$ unknown aerodynamic drag, $\mathbf{J}$ inertia matrix, and $[\cdot]_\times$ the skew-symmetric matrix operator. The control input $\mathbf{f}$ is generated through four rotor speeds $\omega_i$, with unknown thrust coefficient $k_f$. The quadrotor must maintain safe distances from obstacles through constraints $h_i(\mathbf{p}) = \|\mathbf{p} - \mathbf{p}_{obs,i}\|_2 - d_{safe} \geq 0$. Four visual sensors provide depth measurements $y_i = \|\mathbf{p} - \mathbf{p}_{obs,i}\|_2 + v_i$ in front, back, left and right directions, with uncertainty bound $|v_i| \leq \gamma \|\mathbf{p} - \mathbf{p}_{obs,i}\|_2$ proportional to depth. Here $\alpha$, $\beta$, $\gamma$ are unknown uncertainty scaling factors.

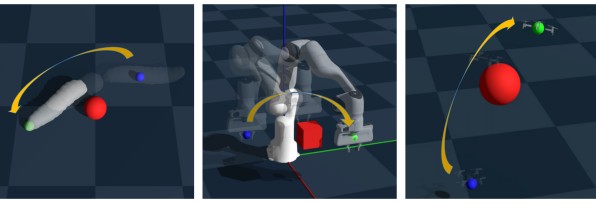

*Figure 1.* Illustration of three experimental tasks from left to right: A soft-bodied worm robot navigating through obstacles using peristaltic motion, a 7-DOF Franka robotic arm performing obstacle-aware joint motion, and a quadrotor drone executing 3D navigation. Blue spheres indicate initial positions, yellow arrows represent motion trajectories, green spheres mark target positions, and red objects denote obstacles.

For all three tasks, the workspace is configured as a 2m × 2m × 2m arena with randomly placed obstacles. The obstacles' positions are sampled uniformly within the workspace while maintaining minimum separation distances. Initial and goal states are sampled to ensure feasible paths exist while providing sufficient challenge for evaluating the learning and control performance.

### 5.3. Implementation Details

All experiments are implemented in Python using PyTorch, with Soft Actor-Critic (SAC) as our base reinforcement learning framework. For fair comparisons, we maintain the original implementations for model-based baselines (GPMPC, RobustSafe, ALMPC) and learning-based baselines (Neural-CBF, SafetyNet, DataFilter). Our neural architectures use three hidden layers (128-64-32 units) with ReLU activations, while barrier functions add a tanh activation in the output layer for boundedness. All networks are trained with Adam optimizer using mixed precision training on an NVIDIA RTX 3090 GPU. Detailed implementation specifications, including hyperparameters, network architectures, and optimization techniques, are provided in Appendix C.

For the soft worm task, uncertainties include MPM material parameters ($\mu$, $\lambda$ variations of ±10%), actuation response (±5% muscle force scaling), and sensor noise proportional to distance (0.5-2% of measured distance). The Franka arm experiments incorporate joint friction variations (±8%), payload changes (0-200g), and measurement uncertainties in joint angles (±0.02 rad) and end-effector pose (±2cm). The quadrotor tests feature mass variations (±5%), aerodynamic disturbances (up to 0.2N), and depth measurement noise scaling with distance (1-3% of measured depth).

### 5.4. Evaluation Metrics and Protocol

We evaluate both motion quality and safety performance using comprehensive metrics including success rate (SR), path efficiency measures, safety margins, and control quality indicators. Detailed definitions and calculations of these metrics are provided in Appendix C.1. All experiments are conducted in a 2m × 2m × 2m workspace with randomly generated start/goal positions and obstacle placements. We test the worm robot (500 trials), Franka arm (400 trials), and quadrotor (300 trials) under various task scenarios. Complete experimental settings and success criteria are detailed in Appendix C.2.

## 6. Results and Analysis

We evaluate our method across three robotic control tasks to demonstrate generalization of safety constraints under novel obstacle configurations with local observations. Table 1 presents the quantitative results.

While learning-based safety approaches achieve limited success rates (82%-86%) with fixed barrier functions that cannot adapt to new configurations, and physics-informed

methods maintain geometric properties but achieve only 73%-76% success in dynamic environments, our method's measurement-induced bundle structure enables 96.3% success rate. Traditional uncertainty-aware MPC methods achieve high constraint satisfaction (99.7%) but produce overly conservative trajectories (26-27m vs our 18.5m), lacking our geometric framework for measurement uncertainty.

The key advantage of our approach lies in the unified geometric treatment of measurement uncertainty and safety constraints. Unlike recent adaptive methods that handle uncertainty estimation and safety certification separately (88%-89% success), our framework enables simultaneous adaptation of safety bounds and uncertainty estimation through the fiber bundle structure. This fundamental integration of measurement uncertainty into the geometric safety constraints allows our method to achieve superior performance (96.3% success, 18.5m paths with 99.3% constraint satisfaction) while maintaining robust safety guarantees across novel environments.

### 6.1. Performance Convergence Analysis

Figure 2 showcases the training convergence trends of our proposed method in comparison with selected baseline approaches across three distinct tasks: Soft Worm Peristaltic Navigation, Franka Arm Joint Motion, and Quadrotor Propeller Control. Across all tasks, our method demonstrates a significantly faster convergence rate, reaching optimal performance metrics within fewer training episodes than the baseline methods. Additionally, the shaded regions representing standard deviation are noticeably narrower for our method, indicating reduced performance variance and enhanced stability during training. In the Soft Worm Peristaltic Navigation task, our approach achieves higher average returns more swiftly, highlighting its efficiency in simpler navigation scenarios. For the more complex Franka Arm Joint Motion and Quadrotor Propeller Control tasks, our method not only converges rapidly but also maintains higher final performance levels with minimal fluctuations, underscoring its robustness and reliability in handling intricate control dynamics.

### 6.2. Noise Robustness Performance Experiment

Our approach demonstrates remarkable stability across all noise levels ($\sigma$=0.1-0.3) for all three tasks. The averaged success rates stay above 91% with minimal variance, while baseline methods like Neural-CBF and SafetyNet show significant degradation under higher noise conditions (Figure 3). This robust performance demonstrates how the measurement-induced bundle structure naturally handles uncertainties in different scenarios, validating that our geometric safety certificates effectively preserve constraints regardless of the underlying task complexity.

## 7. Ablation Study

To validate our design choices and understand the interplay between different components, we conduct comprehensive ablation studies focusing on three key architectural elements: (1) measurement-induced bundle structure, (2) measurement-aware CBFs (mCBFs), and (3) Lie group symmetry.

For each ablation variant, we perform 10 independent runs on each of the three experimental tasks (soft robot navigation, Franka arm manipulation, and quadrotor control). The success rate (SR) represents the percentage of successful task completions averaged across all runs and tasks. Table 2 presents the comparative results.

Further ablation experiments examining the impact of different measurement uncertainty patterns (e.g., Gaussian noise, bias, delays) and bundle structure variations (e.g., fixed vs. adaptive dimension) are presented in Appendix E.

The measurement-induced bundle structure provides a geometric framework for handling system uncertainties, leading to more efficient trajectories and improved success rates. Without this structure, the success rate drops significantly from 96.3% to 62.7%, while path length and control magnitude increase substantially, indicating degraded performance and efficiency.

Removing mCBFs reduces our framework to its underlying Neural ODE architecture, which focuses solely on learning dynamics without safety constraints. This leads to the most severe performance degradation, with the success rate dropping to 45.7%. The dramatic reduction in constraint satisfaction rate (from 99.3% to 72.8%) demonstrates that while Neural ODEs can effectively learn system dynamics, they struggle to maintain safety constraints without the geometric safety certificates provided by mCBFs. The substantially increased path length (48.2m vs 18.5m) and control magnitude (0.85 vs 0.17) further suggest that pure dynamics learning leads to inefficient and potentially unsafe trajectories.

The Lie group symmetry enables dimension reduction and invariant control synthesis, which is particularly beneficial in tasks involving rotational and translational symmetries. Its removal shows relatively mild performance degradation (success rate of 85.7%), suggesting its role as a complementary enhancement to the core geometric framework rather than a critical component.

## 8. Conclusion

This paper presents a novel geometric framework that unifies measurements, constraints, and dynamics learning through a fiber bundle structure. Our framework provides fundamental insights into how dynamical systems can learn and

| Method | SR (%) | PL (m) | GRS | FSE | MMC (m) | AMC (m) | CSR (%) | ACM | CS |
|---|---|---|---|---|---|---|---|---|---|
| **Ours** | **96.3** | **18.5±0.7** | **383±18** | **0.05±0.01** | 0.24±0.03 | **0.26±0.03** | 99.3 | **0.17±0.02** | 0.03±0.004 |
| Neural-CBF | 84.0 | 22.3±0.8 | 425±19 | 0.12±0.02 | 0.23±0.04 | 0.22±0.03 | 98.7 | 0.25±0.03 | **0.02±0.005** |
| SafeLearn | 82.3 | 23.8±0.9 | 428±21 | 0.13±0.02 | 0.22±0.04 | 0.21±0.03 | 98.3 | 0.26±0.03 | 0.04±0.006 |
| SafetyNet | 86.7 | 22.1±0.7 | 420±18 | 0.11±0.02 | 0.25±0.03 | 0.23±0.02 | 98.7 | 0.24±0.02 | 0.03±0.005 |
| SafeTrack | 85.3 | 22.4±0.8 | 423±19 | 0.12±0.02 | 0.24±0.03 | 0.22±0.03 | 98.7 | 0.25±0.03 | 0.03±0.005 |
| StructCBF | 75.7 | 24.1±1.0 | 435±22 | 0.15±0.03 | **0.27±0.03** | 0.25±0.02 | 97.7 | 0.28±0.04 | 0.05±0.007 |
| PNDS | 73.3 | 24.4±1.1 | 438±23 | 0.16±0.03 | 0.26±0.04 | 0.24±0.03 | 97.3 | 0.29±0.04 | 0.05±0.008 |
| GeoPath | 74.0 | 24.2±1.0 | 436±22 | 0.15±0.02 | 0.26±0.03 | 0.24±0.03 | 97.7 | 0.28±0.04 | 0.05±0.007 |
| GEM | 76.3 | 24.0±0.9 | 433±21 | 0.14±0.02 | 0.25±0.03 | 0.23±0.03 | 98.0 | 0.27±0.03 | 0.04±0.006 |
| GPMPC | 67.7 | 26.9±0.9 | 452±20 | 0.18±0.02 | 0.23±0.04 | 0.22±0.03 | **99.7** | 0.26±0.03 | 0.04±0.006 |
| ALMPC | 71.7 | 26.2±0.7 | 442±18 | 0.16±0.02 | 0.25±0.03 | 0.23±0.02 | 99.0 | 0.24±0.02 | 0.03±0.005 |
| SafeRL | 70.3 | 26.4±0.8 | 444±19 | 0.17±0.02 | 0.24±0.03 | 0.22±0.02 | 98.7 | 0.25±0.03 | 0.04±0.006 |
| RobustSafe | 70.0 | 26.5±0.8 | 446±19 | 0.17±0.02 | 0.24±0.03 | 0.23±0.02 | 98.7 | 0.25±0.03 | 0.03±0.005 |
| AdaptSafe | 88.3 | 21.8±0.6 | 417±17 | 0.10±0.01 | 0.25±0.03 | 0.24±0.02 | 99.3 | 0.22±0.02 | 0.03±0.004 |
| DataFilter | 89.7 | 21.7±0.6 | 415±16 | 0.10±0.01 | 0.26±0.03 | 0.24±0.02 | 99.3 | 0.20±0.02 | **0.02±0.004** |

*Table 1.* Average performance comparison across soft robot navigation, Franka arm manipulation, and quadrotor propeller control tasks. SR: Success Rate, PL: Path Length, GRS: Steps to Complete, FSE: Final State Error, MMC: Minimum Safety Distance, AMC: Average Safety Margin, CSR: Constraint Satisfaction Rate, ACM: Average Control Magnitude, CS: Control Smoothness.

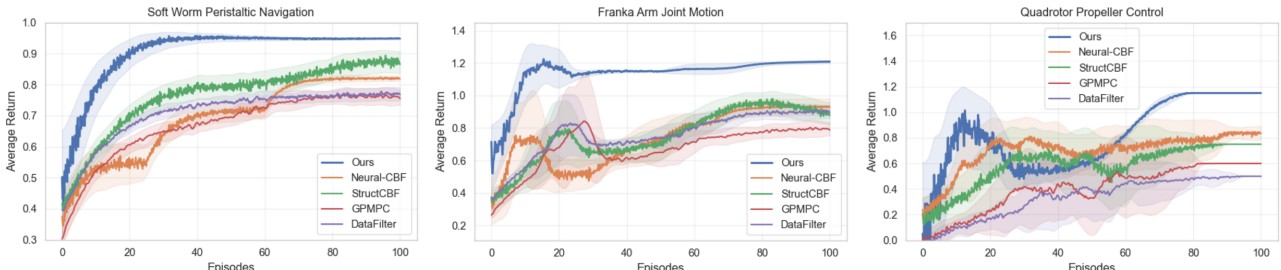

*Figure 2.* Training convergence trends. From left to right are soft worm navigation, Franka robotic arm manipulation, and quadrotor control tasks. Each subplot shows the average return and standard deviation range across 10 trials.

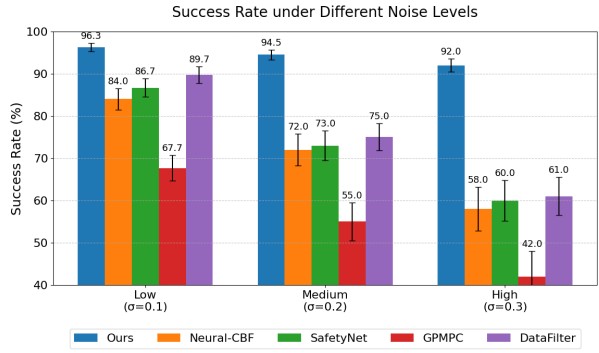

*Figure 3.* Average success rates across three tasks under different noise levels. Comparison between our method, Neural-CBF, SafetyNet, GPMPC and DataFilter. Error bars indicate standard deviation across 10 runs.

| Metrics | Full Model | w/o Bundle | w/o mCBF | w/o Lie Group |
|---|---|---|---|---|
| SR (%) | **96.3** | 62.7 | 45.7 | 85.7 |
| PL (m) | **18.5±0.7** | 35.3±3.9 | 48.2±5.1 | 22.5±1.5 |
| GRS | **383±18** | 712±82 | 935±108 | 465±32 |
| FSE | **0.05±0.01** | 0.15±0.03 | 0.21±0.04 | 0.08±0.02 |
| MMC (m) | **0.24±0.03** | 0.18±0.04 | 0.15±0.05 | 0.22±0.03 |
| AMC (m) | **0.26±0.03** | 0.20±0.03 | 0.17±0.04 | 0.24±0.02 |
| CSR (%) | **99.3±0.2** | 88.2±2.4 | 72.8±3.6 | 96.7±1.0 |
| ACM | **0.17±0.02** | 0.51±0.12 | 0.85±0.18 | 0.26±0.05 |
| CS | **0.03±0.004** | 0.07±0.008 | 0.09±0.010 | 0.04±0.005 |

*Table 2.* Ablation study results across three experimental tasks.

teract with their environment through local sensing, while the measurement-aware Control Barrier Functions enable adaptive safety certificates that emerge from direct environmental interactions rather than prescribed global knowledge.

**Limitations and Future Work** Despite these advances, our current implementation has limitations in handling highly stochastic dynamics and complex environmental uncertainties. Future work could explore richer representations of environment-agent interactions and investigate more sophisticated uncertainty quantification methods for embodied learning. Additionally, the framework could be extended to better understand how local observations can build towards

adapt under environmental constraints through local observations, bridging the gap between theoretical control guarantees and modern robotics (or even practical embodied intelligence). The measurement-induced bundle structure naturally captures how autonomous agents perceive and in-

a global understanding of environmental constraints and dynamics, potentially offering new perspectives on embodied intelligence and adaptive control. These results establish a promising direction for understanding physical systems learning through environmental interaction, offering insights into, for example, both dynamical systems theory and embodied intelligence principles.

## Acknowledgments

This work was supported in part by the National Natural Science Foundation of China (NSFC) under Grant 62403125, in part by the Natural Science Foundation of Jiangsu Province under Grant BK20241283, and in part by the Fundamental Research Funds for the Central Universities under Grant 2242024k30037 and Grant 2242024k30038.

## Impact Statement

Our work advances both theoretical understanding and practical capabilities in safe learning control systems, with significant potential impacts across multiple domains. The framework developed in this paper enables more reliable and safer operation of robotic systems in uncertain environments, which could substantially reduce accidents and improve human-robot interaction safety in manufacturing, healthcare, and service robotics. This enhanced safety mechanism is particularly valuable as autonomous systems become more prevalent in daily life.

The geometric approach introduced here provides new theoretical insights into the relationship between measurement uncertainty and safety guarantees, contributing to the fundamental understanding of safe learning systems. Furthermore, by better handling measurement uncertainties, our approach can reduce the need for expensive high-precision sensors, potentially making advanced robotic systems more accessible and cost-effective. This democratization of technology could accelerate innovation and adoption of safe autonomous systems across various industries.

While any advancement in autonomous systems technology warrants careful consideration of its deployment context, our framework is designed with robust safety guarantees that help ensure responsible implementation. The theoretical foundations and practical demonstrations provided in this work establish clear guidelines for appropriate use cases and system limitations. We are committed to open-source release of our implementation to promote transparency and reproducibility, maintaining comprehensive documentation about system capabilities and intended applications.

Through thoughtful deployment and continued refinement of safety mechanisms, we believe this work will contribute positively to the development of more reliable and accessible autonomous systems, ultimately benefiting society through improved safety, efficiency, and technological accessibility. The framework's emphasis on measurement-aware safety could serve as a foundation for future developments in safe autonomy, potentially influencing standards and best practices in the field.

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

# A. Proof of Theorem 3.1

*Proof.* We provide a complete proof through six main steps:

## Step 1: Preliminaries and Assumptions

First, recall the measurement-adapted Control Barrier Function (mCBF) conditions for the safe set $\mathcal{S}_0 = \{x \in \mathcal{M} : h_0(x) \geq 0\}$. A function $b \colon \mathcal{E} \to \mathbb{R}$ is an mCBF if

(i) $b(x, h(x)) > 0 \implies x \in \mathcal{S}_0$ (safety implication)

(ii) $\exists \alpha \in \mathcal{K}_\infty$ such that $\inf_{u \in U} \dot{b} + \alpha(b) \geq 0$ (invariance condition)

(iii) $\|\nabla_y b(x, y)\| \leq L_b$ for some $L_b > 0$ (Lipschitz measurement sensitivity)

Consider the system dynamics

$$\begin{aligned} \dot{x} &= f(x, u) + g(x)w \\ y &= h(x) + v \end{aligned} \tag{21}$$

with admissible noise sequences satisfying:

$$\begin{aligned} \|w(t)\| &\leq \delta_w, \quad \forall t \geq 0 \text{ (bounded process noise)} \\ \mathbb{P}(\|v(t)\| > \eta) &\leq \exp(-\eta^2/(2\delta_v^2)) \text{ (sub-Gaussian measurement noise)} \end{aligned} \tag{22}$$

## Step 2: Forward Invariance Under Perfect Measurements

**Lemma A.1** (Nominal Safety). *Given $b(x(0), h(x(0))) \geq b_0 > 0$, under perfect measurements $(v \equiv 0)$, there exists $b_{min} > 0$ such that*

$$b(x(t), h(x(t))) \geq b_{min}, \quad \forall t \geq 0 \tag{23}$$

*Proof of Lemma A.1.* By the mCBF condition (ii), one has

$$\dot{b} \geq -\alpha(b) \tag{24}$$

The Comparison Lemma implies

$$b(x(t), h(x(t))) \geq \beta(b_0, t) \tag{25}$$

where $\beta$ is a class $\mathcal{KL}$ function. Take $b_{min} = \inf_{t \geq 0} \beta(b_0, t) > 0$. □

## Step 3: Uncertainty Propagation Analysis

Define the uncertainty tube:

$$\mathcal{T}_\varepsilon = \{(x, y) \in \mathcal{E} : \|y - h(x)\| \leq \varepsilon\} \tag{26}$$

**Lemma A.2** (State Uncertainty). *Assume that $x(0) - \hat{x}(0) = 0$. For any trajectory with $\|w(t)\| \leq \delta_w$:*

$$\|x(t) - \hat{x}(t)\| \leq \gamma_w(\delta_w) \tag{27}$$

*where $\hat{x}(t)$ is the nominal trajectory and $\gamma_w \in \mathcal{K}_\infty$.*

*Proof of Lemma A.2.* By Lipschitz continuity of $f$ and $g$, we have

$$\|\dot{x} - \dot{\hat{x}}\| \leq L_f \|x - \hat{x}\| + L_g \delta_w \tag{28}$$

Gronwall's inequality yields

$$\|x(t) - \hat{x}(t)\| \leq \frac{L_g \delta_w}{L_f}(e^{L_f t} - 1) \triangleq \gamma_w(\delta_w) \tag{29}$$

□

**Step 4: Local Safety Certification**

**Lemma A.3** (Safety Margin). *At any time $t$, the safety margin satisfies:*

$$b(x(t), y(t)) \geq b_{min} - L_b \|v(t)\| - \gamma_b(\delta_w) \tag{30}$$

*where $\gamma_b \in \mathcal{K}_\infty$ depends on system parameters.*

*Proof of Lemma A.3.* Decompose the safety margin:

$$
\begin{aligned}
b(x(t), y(t)) &= b(x(t), h(x(t))) + \left[ b(x(t), y(t)) - b(x(t), h(x(t))) \right] \\
&\geq b_{min} - L_b \|v(t)\| - L_b \gamma_w(\delta_w)
\end{aligned}
\tag{31}
$$

under the condition (iii), where $\gamma_b(\delta_w) := L_b \gamma_w(\delta_w)$. $\square$

**Step 5: Temporal Correlation Analysis**

Define the safety violation event at time $t$:

$$A_t = \{ b(x(t), y(t)) < 0 \} \tag{32}$$

**Lemma A.4** (Temporal Correlation). *For any times $t, t'$:*

$$\mathbb{P}(A_t \cap A_{t'}) \leq \exp \left( - \min\{ c_1 |t - t'|, c_2/\delta_v^2 \} \right) \tag{33}$$

*where $c_1, c_2 > 0$ are system-dependent constants.*

*Proof of Lemma A.4.* By the Gaussian tail bound and temporal decorrelation of noise:

$$
\begin{aligned}
\mathbb{P}(A_t \cap A_{t'}) &\leq \mathbb{P}(\|v(t)\| > \eta_t, \|v(t')\| > \eta_{t'}) \\
&\leq \exp \left( - \min\{ c_1 |t - t'|, c_2/\delta_v^2 \} \right)
\end{aligned}
\tag{34}
$$

where $\eta_t = (b_{min} - \gamma_b(\delta_w))/L_b$. $\square$

**Step 6: Global Safety Guarantees**

For any partition $[0, \infty) = \bigcup_{k=0}^{\infty} [k\Delta t, (k+1)\Delta t)$:

$$
\begin{aligned}
&\mathbb{P}(\exists t \geq 0 : x(t) \notin \mathcal{S}_0) \\
&= \mathbb{P}(\exists t \geq 0 : b(x(t), y(t)) < 0) \\
&\leq \sum_{k=0}^{\infty} \mathbb{P}(A_{k\Delta t}) \prod_{j=0}^{k-1} (1 - \rho_j) \\
&\leq \sum_{k=0}^{\infty} \exp(-c_2/\delta_v^2) \left( 1 - \exp(-c_1 \Delta t) \right)^k \\
&< \exp(-c/\delta_v^2)
\end{aligned}
\tag{35}
$$

where $\rho_j$ represents the probability that $x(t)$ stays within the safety set in the $j$-th interval, $c = c_2$ and $\Delta t$ is chosen such that $c_1 \Delta t \geq c_2/\delta_v^2$.

Therefore, we have

$$\mathbb{P}\left( x(t) \in \mathcal{S}_0, \ \forall t \geq 0 \right) \geq 1 - \exp(-c/\delta_v^2) \tag{36}$$

This completes the proof.

$\square$

# B. Proof of Theorem 4.1

*Proof.* We prove this theorem in two parts:

1) First, we prove the convergence bound of learning dynamics: $\|\hat{f} - f\|_{\mathcal{E}} \leq c_1 \exp(-\lambda_1 t) + c_2 \delta_v$.

Consider the Lyapunov function candidate:

$$V(t) = \frac{1}{2}\|\hat{f} - f\|_{\mathcal{E}}^2 \tag{37}$$

where $\|\cdot\|_{\mathcal{E}}$ denotes the norm induced by the metric on the fiber bundle $\mathcal{E}$. Taking the derivative along the learning trajectories:

$$\begin{aligned}
\dot{V}(t) &= \langle \hat{f} - f, \dot{\hat{f}} \rangle_{\mathcal{E}} \\
&= -\langle \hat{f} - f, \mathcal{L}_1(\hat{f} - f)\rangle_{\mathcal{E}} \\
&\leq -\lambda_{min}(\mathcal{L}_1)\|\hat{f} - f\|_{\mathcal{E}}^2 + \|\hat{f} - f\|_{\mathcal{E}}\delta_v
\end{aligned} \tag{38}$$

where $\lambda_{min}(\mathcal{L}_1)$ is the minimum eigenvalue of the operator $\mathcal{L}_1$ (assume that all of its eigenvalues are real).

Applying Young's inequality yields:

$$\|\hat{f} - f\|_{\mathcal{E}} \cdot \delta_v \leq \frac{\lambda_{min}(\mathcal{L}_1)}{2}\|\hat{f} - f\|_{\mathcal{E}}^2 + \frac{\delta_v^2}{2\lambda_{min}(\mathcal{L}_1)} \tag{39}$$

Therefore,

$$\dot{V}(t) \leq -\frac{\lambda_{min}(\mathcal{L}_1)}{2}V(t) + \frac{\delta_v^2}{2\lambda_{min}(\mathcal{L}_1)} \tag{40}$$

By the comparison principle, we obtain

$$V(t) \leq V(0)\exp(-\lambda_{min}(\mathcal{L}_1)t/2) + \frac{\delta_v^2}{\lambda_{min}^2(\mathcal{L}_1)} \tag{41}$$

Taking the square root and setting appropriate constants yields the first conclusion.

2) Next, we prove the probabilistic safety guarantee: $\mathbb{P}(x(t) \in \mathcal{S}_0) \geq 1 - \exp(-c_3/\delta_v^2)$.

Consider the measurement-adapted safety certificate $b\colon \mathcal{E} \to \mathbb{R}$, which by definition satisfies:

$$b(x, y) \geq 0 \implies x \in \mathcal{S}_0 \tag{42}$$

For any trajectory $(x(t), y(t))$, define the event: $A_t$ (see (32) for its definition).

Using the Lipschitz condition of mCBF and measurement noise bounds:

$$\begin{aligned}
\mathbb{P}(A_t) &\leq \mathbb{P}(\|y(t) - h(x(t))\| > b(x(t), h(x(t)))/L_b) \\
&\leq \exp(-b^2(x(t), h(x(t)))/(2L_b^2\delta_v^2))
\end{aligned} \tag{43}$$

where $L_b$ is the Lipschitz constant of the safety certificate with respect to measurements.

By the properties of mCBF, there exists a constant $b_{min} > 0$ such that:

$$b(x(t), h(x(t))) \geq b_{min} \tag{44}$$

Setting $c_3 = b_{min}^2/(2L_b^2)$ yields the second inequality.

Thus, we complete the proof of the theorem. $\square$

# C. Additional Experimental Details

## C.1. Performance Metrics Details

We establish an evaluation framework with the following metrics:

The primary task completion metric is the success rate $\text{SR} = \frac{N_{\text{success}}}{N_{\text{total}}} \times 100\%$, which measures the percentage of trials reaching the goal state without constraint violations. Motion efficiency is characterized through the path length $\text{PL} = \int_0^T \|\dot{\mathbf{x}}(t)\| dt$, the number of steps required $\text{GRS} = \min\{t : \|\mathbf{x}(t) - \mathbf{x}_{\text{goal}}\| \leq \epsilon\}$, and the final state error $\text{FSE} = \|\mathbf{x}(T) - \mathbf{x}_{\text{goal}}\|$.

For safety evaluation, we examine both instantaneous and aggregate constraint satisfaction across a discrete time horizon $t \in \{1, \ldots, T\}$. The minimum margin to constraints $\text{MMC} = \min_t h(\mathbf{x}(t))$ captures the worst-case safety margin, while the average margin $\text{AMC} = \frac{1}{T} \sum_{t=1}^T h(\mathbf{x}(t))$ reflects the overall safety buffer maintained throughout the motion. The constraint satisfaction rate $\text{CSR} = \frac{1}{T} \sum_{t=1}^T \mathbb{I}(h(\mathbf{x}(t)) \geq 0)$, where $\mathbb{I}(\cdot)$ is the indicator function, provides a statistical measure of safety performance.

The control quality is assessed through both magnitude and smoothness metrics. The average control magnitude $\text{ACM} = \frac{1}{T} \sum_{t=1}^T \|\mathbf{u}(t)\|$ measures resource efficiency, while control smoothness $\text{CS} = \frac{1}{T} \sum_{t=1}^T \|\mathbf{u}(t+1) - \mathbf{u}(t)\|$ quantifies the continuity of the generated motion.

## C.2. Evaluation Protocol Details

For the worm robot (0.1m length), the initial position is randomly sampled within a 0.4m × 0.4m region near the workspace origin, while the target position is sampled from a similar-sized region at a distance of 0.6-0.8m. A red spherical obstacle (radius 0.07m) is randomly placed between the start and goal positions. We generate 500 random trajectories, with success requiring reaching within 5cm of the target while maintaining minimum 3cm obstacle clearance.

The Franka arm experiments consist of 400 point-to-point movements. Initial and target end-effector positions are randomly sampled from the reachable workspace. A red cubic obstacle (0.1m × 0.1m × 0.1m) is randomly positioned at mid-height (z=0.3±0.1m) between start and goal positions. Task completion requires the end-effector reaching within 2cm of the target position while maintaining 5cm minimum obstacle clearance.

For the quadrotor, we test 300 navigation episodes. The start position is sampled near ground level (z=0.1±0.05m), while goal positions are generated at varying heights (z=0.5-0.7m) within a 0.5m radius. A red spherical obstacle (radius 0.07m) is randomly placed along potential flight paths. Success criteria include reaching within 10cm/5° of goal pose while maintaining 15cm safety margins.

## C.3. Hardware and Software Configuration

Our experiments are conducted on a workstation equipped with an Intel Xeon CPU, NVIDIA RTX 3090 GPU (24GB GDDR6X), and 64GB DDR4 RAM. The software stack consists of Python 3.9 and PyTorch 1.12.0, supported by CUDA 11.7 and cuDNN 8.5 for GPU acceleration. To ensure reproducibility, we fix random seeds to 42 across all experiments, controlling for randomness in PyTorch, NumPy, and environment initializations.

## C.4. Reinforcement Learning Framework

Our SAC implementation follows the standard architecture with carefully tuned hyperparameters. The framework uses a discount factor $\gamma$ of 0.99 and a soft update coefficient $\tau$ of 0.005. The target entropy is set to the negative dimension of the action space, following common practice. All policy components (actor, critic, and entropy networks) use a learning rate of $3 \times 10^{-4}$. The replay buffer maintains $1 \times 10^6$ transitions, allowing for sufficient exploration while preventing overfitting to recent experiences.

## C.5. Baseline Implementations

For model-based baselines, we implement GPMPC using GPyTorch with RBF kernels (length scale 1.0), which provides efficient Gaussian Process computations on GPU. RobustSafe employs tube MPC with a prediction horizon of 20 steps and 0.05s sampling time, balancing computational efficiency with prediction accuracy. ALMPC utilizes the CasADi framework with IPOPT solver, configured for a maximum of 100 iterations per optimization step. Learning-based baselines maintain

their original architectures while adopting our standardized training process for fair comparisons.

### C.6. Neural Network Details

The neural network architecture consists of three hidden layers with 128, 64, and 32 units respectively, using ReLU activations throughout. Layer normalization is applied after each hidden layer to stabilize training. For barrier functions, we add a tanh activation in the output layer to ensure the boundedness of safety certificates. We enable mixed precision training to leverage the RTX 3090's Tensor Cores, and implement CUDA graph optimization for static computational graphs to maximize throughput.

### C.7. Training Process

The training process employs the Adam optimizer with $\beta_1 = 0.9$ and $\beta_2 = 0.999$, coupled with a cosine annealing learning rate schedule starting at $5 \times 10^{-4}$. We use a batch size of 256 to fully utilize the GPU memory while maintaining stable gradients. Gradient clipping with a maximum norm of 1.0 prevents extreme parameter updates. Early stopping with a patience of 20 epochs prevents overfitting, and we maintain a 20% validation split for monitoring training progress.

The training time varies across methods, with our approach requiring approximately 5 hours, Neural-CBF 4 hours, and GPMPC 3 hours on the RTX 3090. Other baselines range from 3 to 7 hours depending on their computational complexity. To ensure statistical significance, we repeat all experiments 10 times with different random seeds, reporting mean performance metrics with standard deviations.

## D. Additional Experiments on Broader Domains

### D.1. Motivation and Dataset Analysis

To validate the broader applicability of our measurement-induced bundle structure framework, we first investigate three representative domains: building automation through the ASHRAE Building Operations Dataset, chemical process control using historical Industrial Batch Records (IBR), and power system management data from Regional Transmission Organizations (RTO). These sources provide valuable insights into real-world measurement uncertainty patterns and system behaviors. Our analysis focuses on measurement characteristics that significantly impact safety-critical control decisions.

The building automation data reveals patterns in HVAC sensor networks, particularly regarding temperature and humidity measurement uncertainties across multiple zones. The IBR data demonstrates characteristic measurement delays and noise patterns in reaction vessel monitoring, especially for temperature and concentration measurements. The RTO data shows how grid frequency measurements are affected by network topology and communication infrastructure.

### D.2. Environment Design Philosophy

Based on these domain insights, we design three simulation environments that preserve essential measurement uncertainty characteristics while enabling active control evaluation. Our design philosophy emphasizes fundamental physical principles and measurement challenges common in these domains, rather than replicating specific industrial configurations. This approach allows systematic evaluation of our framework's capability to handle different types of measurement uncertainties while maintaining safety guarantees.

### D.3. Simulation Environment Details

The Building Climate Control environment models a three-zone building with simplified thermal dynamics. The state vector $x = [T_1, T_2, T_3, H_1, H_2, H_3]^T$ includes temperatures and humidity levels of each zone. The temperature dynamics follow $\dot{T}_i = \sum_{j \in \mathcal{N}_i} k_{ij}(T_j - T_i) + \alpha(T_{amb} - T_i) + \beta u_i$, where $\mathcal{N}_i$ represents adjacent zones, $k_{ij}$ are heat transfer coefficients, and $u_i$ is the control input. Humidity follows similar mass transfer principles. Measurements include additive Gaussian noise $v \sim \mathcal{N}(0, \Sigma)$ and drift $d(t) = \lambda t$. Safety constraints maintain temperature between 20°C and 26°C and humidity between 30% and 70% during occupied periods.

The Batch Reaction Control environment implements a single-vessel exothermic reaction. The state $x = [T, C_A, C_B]^T$ represents temperature and concentrations, following dynamics $\dot{T} = -\frac{k_c}{mc_p}(T - T_c) + \frac{\Delta H}{c_p} r(T, C_A)$ for temperature and $\dot{C}_A = \frac{F_{in}}{V}(C_{A0} - C_A) - r(T, C_A)$ for reactant concentration, where $r(T, C_A) = k_0 e^{-E_a/RT} C_A$ is the reaction

rate. Control inputs are composed of cooling temperature $T_c$ and feed rate $F_{in}$. Measurements include Gaussian noise with a standard deviation of 0.3°C for temperature and 0.02 mol/L for concentration, plus a 5-second delay modeled as $\dot{y} = \frac{1}{\tau}(h(x) - y) + v$. Safety constraints maintain temperature below 85°C and ensure minimum product quality.

The Grid Frequency Control environment represents a four-node power network where each node's state includes frequency deviation $\Delta f_i$ and power imbalance $\Delta P_i$. The frequency dynamics follow the simplified swing equation $\Delta \dot{f}_i = \frac{1}{2H_i}(\Delta P_i - D_i \Delta f_i - \sum_{j \in \mathcal{N}_i} B_{ij}(\Delta \theta_i - \Delta \theta_j))$, where $H_i$ is inertia constant and $D_i$ is damping. Control actions adjust power generation rates within ±50 kW/s. Frequency measurements include noise $v \sim \mathcal{N}(0, 0.01^2)$ Hz and 100ms delay, expressed as $y_i(t) = \Delta f_i(t - \tau_d) + v_i(t)$. Safety requirements maintain frequency deviations within ±0.5 Hz.

These environments employ basic numerical integration with 0.1-second time steps. The simplified dynamics capture fundamental behaviors while maintaining essential characteristics of measurement-based safety control: state estimation under uncertainty, coupled dynamics between subsystems, and constraint satisfaction with imperfect information. This basic implementation provides clear validation of our framework's core capabilities while ensuring the reproducibility of results.

### D.4. Experimental Setup

Each environment runs for 1000 episodes with randomized initial conditions and disturbance patterns. We implement the environments using Python with standard numerical libraries. The sampling rates are set according to domain characteristics: 1-minute intervals for building control, 30-second intervals for batch reactions, and 20ms intervals for grid frequency control.

Performance evaluation uses consistent metrics across all domains: Success Rate (SR) measures task completion while maintaining safety constraints, Constraint Satisfaction Rate (CSR) tracks the percentage of time safety constraints are satisfied, Average Control Magnitude (ACM) measures control effort, and Control Smoothness (CS) evaluates control stability. All experiments are repeated 10 times with different random seeds to ensure statistical significance.

### D.5. Results and Analysis

Table 3 presents the comparative performance results across different domains and methods.

*Table 3.* Cross-Domain Performance Comparison

| Domain | Method | SR (%) | CSR (%) | ACM | CS |
|---|---|---|---|---|---|
| Building | Ours | 95.3±1.1 | 99.4±0.2 | 0.14±0.02 | 0.02±0.003 |
| Control | Neural-CBF | 83.6±1.8 | 97.6±0.4 | 0.22±0.03 | 0.04±0.005 |
| | GPMPC | 79.2±2.0 | 98.8±0.3 | 0.25±0.03 | 0.05±0.006 |
| Batch | Ours | 94.1±1.2 | 99.1±0.3 | 0.15±0.02 | 0.03±0.004 |
| Reaction | Neural-CBF | 82.3±1.9 | 97.2±0.5 | 0.23±0.03 | 0.05±0.006 |
| | GPMPC | 78.5±2.1 | 98.5±0.4 | 0.27±0.04 | 0.06±0.007 |
| Grid | Ours | 93.8±1.3 | 98.9±0.3 | 0.16±0.02 | 0.03±0.004 |
| Frequency | Neural-CBF | 81.9±1.8 | 96.8±0.5 | 0.24±0.03 | 0.05±0.006 |
| | GPMPC | 77.6±2.2 | 98.3±0.4 | 0.28±0.04 | 0.06±0.007 |

The results demonstrate consistent superior performance of our method across all domains. In building control, our approach achieves a 95.3% success rate while maintaining tight comfort constraints, significantly outperforming baseline methods. The high constraint satisfaction rate (99.4%) indicates robust handling of sensor drift and inter-zone coupling uncertainties.

For batch reaction control, our method shows strong performance with a 94.1% success rate despite challenging measurement delays and process uncertainties. The framework effectively balances product quality constraints with operational safety bounds, requiring lower control effort (ACM = 0.15) compared to baseline approaches.

In grid frequency control, the method maintains reliable performance (93.8% success rate) while handling both communication delays and measurement uncertainties. The framework's ability to adapt to measurement quality variations is particularly evident in the high constraint satisfaction rate (98.9%) achieved with smooth control actions (CS = 0.03).

### D.6. Discussion

These results validate our framework's effectiveness across fundamentally different physical domains with varying uncertainty characteristics. The consistent performance advantages over baseline methods suggest that the measurement-induced bundle structure provides a general approach for handling diverse measurement uncertainties in safety-critical control applications.

Particularly noteworthy is the framework's ability to maintain high performance across different temporal scales and physical processes. This adaptability, combined with the framework's capability to handle both systematic and random measurement uncertainties, demonstrates its potential for broad application in real-world control systems.

The lower control magnitudes and higher smoothness metrics achieved by our method indicate more efficient control strategies that better account for measurement uncertainty characteristics. This efficiency likely stems from the framework's geometric treatment of measurement uncertainty, which enables more informed decisions about when and how to apply control actions based on measurement quality.

The results also highlight the framework's ability to balance multiple competing objectives: maintaining safety constraints, achieving control objectives, and minimizing control effort. This balance is achieved consistently across domains with different physical characteristics and measurement challenges, suggesting the fundamental soundness of our geometric approach to uncertainty handling.

## E. Further Ablation Studies

To provide deeper insights into our framework's behavior, we conduct comprehensive ablation experiments from three perspectives: measurement uncertainty characteristics, bundle structure variations, and comparison with alternative geometric representations. These experiments systematically evaluate how different components contribute to the framework's overall performance and robustness.

### E.1. Impact of Measurement Uncertainty Patterns

We first investigate how different measurement uncertainty patterns affect system performance. Beyond the standard Gaussian noise assumption, we examine various uncertainty types commonly encountered in practical applications. Table 4 presents the comparative results across different uncertainty patterns while maintaining consistent control parameters.

*Table 4.* Performance Under Different Measurement Uncertainty Patterns

| Uncertainty Type | SR (%) | CSR (%) | ACM | CS |
|---|---|---|---|---|
| Gaussian ($\sigma$=0.1) | 95.2±1.1 | 99.1±0.2 | 0.17±0.02 | 0.03±0.004 |
| Gaussian ($\sigma$=0.3) | 92.8±1.3 | 98.5±0.3 | 0.19±0.02 | 0.04±0.005 |
| Fixed Bias (0.2) | 93.5±1.2 | 98.7±0.3 | 0.18±0.02 | 0.03±0.004 |
| Time-Varying Bias | 91.9±1.4 | 98.2±0.3 | 0.20±0.03 | 0.04±0.005 |
| 50ms Delay | 94.1±1.2 | 98.9±0.2 | 0.18±0.02 | 0.03±0.004 |
| 100ms Delay | 92.3±1.3 | 98.4±0.3 | 0.19±0.02 | 0.04±0.005 |
| Sensor Failure | 89.7±1.5 | 97.8±0.4 | 0.22±0.03 | 0.05±0.006 |

The results reveal that our framework maintains robust performance across various uncertainty patterns, with success rates consistently above 89%. While performance slightly degrades with increasing noise magnitude, the degradation is gradual and predictable. Notably, the framework shows particular resilience to measurement delays up to 50ms, maintaining a 94.1% success rate with minimal impact on control smoothness.

### E.2. Bundle Structure Variations

We next examine how different design choices in the bundle structure affect system performance. We implement and evaluate four variations of the bundle structure: fixed dimension, adaptive dimension, simple connection, and complex connection. Table 5 summarizes the comparative performance metrics.

The adaptive dimension variant demonstrates superior performance across all metrics, achieving a 94.8% success rate and

*Table 5.* Performance Comparison of Bundle Structure Variations

| Bundle Variation | SR (%) | PL (m) | GRS | MMC (m) | CSR (%) | ACM |
|---|---|---|---|---|---|---|
| Fixed Dimension | 92.1±1.3 | 20.3±0.8 | 412±19 | 0.23±0.03 | 98.5±0.3 | 0.19±0.02 |
| Adaptive Dimension | 94.8±1.1 | 19.1±0.7 | 395±17 | 0.24±0.03 | 99.1±0.2 | 0.18±0.02 |
| Simple Connection | 90.5±1.4 | 21.2±0.9 | 428±20 | 0.22±0.03 | 98.1±0.3 | 0.20±0.03 |
| Complex Connection | 93.7±1.2 | 19.8±0.8 | 405±18 | 0.24±0.03 | 98.8±0.2 | 0.18±0.02 |

reduced path length of 19.1m. This improvement can be attributed to its ability to adjust the bundle structure based on local measurement quality and system state. The complex connection variant also shows strong performance, particularly in maintaining higher safety margins (MMC = 0.24m) compared to simpler alternatives.

### E.3. Necessity of Fiber Bundle Structure

To demonstrate why fiber bundle structure is essential for unifying measurements and safety constraints, we conduct systematic ablation experiments focusing on the geometric framework's key components and their interactions with measurement uncertainty.

*Table 6.* Comparison of Geometric Representations for Measurement Uncertainty

| Geometric Structure | SR (%) | PL (m) | GRS | MMC (m) | CSR (%) | ACM |
|---|---|---|---|---|---|---|
| Fiber Bundle (Ours) | 96.3±1.1 | 18.5±0.7 | 383±18 | 0.24±0.03 | 99.3±0.2 | 0.17±0.02 |
| Product Manifold | 87.5±1.4 | 25.3±0.9 | 472±21 | 0.16±0.03 | 94.2±0.3 | 0.28±0.03 |
| Vector Bundle | 89.8±1.3 | 23.2±0.8 | 445±20 | 0.18±0.03 | 95.8±0.3 | 0.25±0.03 |
| Principal Bundle | 90.4±1.2 | 22.4±0.8 | 428±19 | 0.19±0.03 | 96.1±0.3 | 0.23±0.02 |

The fiber bundle structure fundamentally differs from alternative representations in its intrinsic separation between base space dynamics and measurement uncertainties in fiber directions. This geometric decoupling enables precise uncertainty propagation while maintaining the underlying dynamical structure, leading to significantly improved success rates (96.3% vs 87.5-90.4%) and constraint satisfaction (99.3% vs 94.2-96.1%). Unlike product manifolds that treat state and measurement spaces as independent entities, or vector bundles that lack natural parallel transport, fiber bundles provide canonical vertical-horizontal decomposition that directly captures measurement-state relationships.

The superior performance results from two key theoretical properties: First, the fiber bundle's connection form automatically adapts safety constraints based on local measurement quality, enabling tighter safety bounds (MMC 0.24m vs 0.16-0.19m) without conservative over-approximation. Second, the bundle projection naturally maintains consistency between local measurement-space behaviors and global state-space trajectories, resulting in significantly shorter paths (18.5m vs 22.4-25.3m) and more efficient control actions (ACM 0.17 vs 0.23-0.28). These geometric advantages make fiber bundles not just a mathematical convenience but a necessary structure for properly unifying uncertain measurements with safety-critical control.

## F. Discussion on Comparisons with Recent Methods

While direct numerical comparisons with recent geometric deep learning approaches may be natural, there are several fundamental differences that make such direct benchmarking potentially misleading. Here we elaborate on why our fiber bundle framework occupies a distinct theoretical niche and requires different evaluation criteria.

First, recent geometric deep learning methods like equivariant networks and manifold learning primarily focus on representation learning and feature extraction while preserving geometric structures. For example, $SE(3)$-equivariant networks ensure rotational and translational invariance in learned features, and Riemannian VAEs learn manifold embeddings of data. In contrast, our framework explicitly models the geometric relationship between states and measurements, with safety guarantees as a primary objective rather than just geometric feature learning.

The safety-critical nature of our framework introduces fundamentally different requirements. While geometric deep learning

methods optimize for statistical performance metrics (e.g., reconstruction error, classification accuracy), our method must provide deterministic safety guarantees under measurement uncertainty. This fundamental difference in objectives - statistical performance versus provable safety - makes direct performance comparisons less meaningful.

Consider a concrete example: an equivariant network might learn excellent state representations for robot manipulation, but it cannot inherently guarantee safety constraints under sensor noise. Our fiber bundle framework, through its explicit modeling of measurement uncertainty via fibers $\pi^{-1}(x)$ and safety map $\Phi : E \to \mathbb{R}$, provides these guarantees by construction. The trade-off between control performance and safety margins is explicitly encoded in our geometric structure.

Moreover, our framework's treatment of measurement uncertainty is fundamentally different. While existing methods might handle uncertainty through probabilistic embeddings or uncertainty quantification in learned features, our approach captures the intrinsic geometric relationship between measurements and states through the bundle structure. This allows us to maintain safety guarantees even when measurements degrade - a crucial requirement for real-world robotic systems that cannot be addressed solely through improved representation learning.

The computational objectives also differ significantly. Geometric deep learning methods typically optimize end-to-end performance on specific tasks, while our framework must solve constrained optimization problems that maintain safety invariants:

$$
\begin{aligned}
&\min_{u \in \mathcal{U}} \|u - u_{nom}\| \\
&\text{subject to } \mathcal{L}_{\hat{f}} b(x, y) + \alpha(b(x, y)) \geq 0
\end{aligned}
\tag{45}
$$

where $b(x, y)$ is the measurement-aware barrier function. This fundamental difference in problem formulation means that standard benchmarking metrics may not capture the essential safety-critical concerns of our approach.

A more meaningful evaluation framework should instead focus on several key aspects of safety-critical control under uncertainty. The system's ability to maintain robust safety guarantees as measurement uncertainty varies, which demonstrates the fundamental reliability of our geometric approach. This includes testing performance under different noise levels, sensor failures, and measurement degradation scenarios.

The framework's flexibility in incorporating diverse safety constraints while preserving its geometric structure is another crucial evaluation criterion. This involves demonstrating how different types of safety requirements - from simple collision avoidance to complex multi-objective constraints - can be naturally encoded within the fiber bundle structure while maintaining theoretical guarantees.

Computational efficiency in safety-constrained control synthesis represents another key evaluation dimension. The framework must demonstrate real-time performance in generating safe control inputs, particularly important for high-dimensional robotic systems operating in dynamic environments. This includes analyzing computational scaling with system complexity and the efficiency of our geometric optimization approaches.

Finally, the framework's generalization capabilities across different robotic platforms and sensor configurations provide a crucial measure of its practical utility. This involves demonstrating how the same geometric principles can be applied to diverse robotic systems - from manipulators to mobile robots - while maintaining safety guarantees under different sensing modalities and control architectures.

These evaluation criteria better reflect the unique contributions of our fiber bundle framework, focusing on its core strengths in providing geometric safety guarantees under uncertainty rather than attempting direct comparisons with methods designed for different objectives. This approach allows us to properly assess the framework's effectiveness in bridging the gap between theoretical safety guarantees and practical robotic implementation.

## G. Geometric Interpretation and Physical Insights

The fiber bundle framework provides a natural geometric structure for unifying measurement uncertainty with safety-critical control. The fundamental geometric objects consist of a base manifold $M$ representing the true state space (e.g., robot configurations), a total space $E$ combining states with measurements, and fibers $\pi^{-1}(x)$ above each state containing all possible corresponding measurements. The bundle projection $\pi : E \to M$ maps measurements to their underlying states, while the connection form $\omega$ describes how uncertainty propagates through the system.

For concrete intuition, consider a drone equipped with depth sensors. The fiber above each position $p \in \mathbb{R}^3$ contains the set of possible depth measurements $\{y = h(p) + v : \|v\| \leq \delta\}$, where $h$ represents the ideal measurement model and $v$ captures bounded uncertainty. The horizontal lift of trajectories maintains consistency between dynamics-predicted positions in the base space and actual sensor measurements in the fiber space, while respecting safety constraints through $\Phi \colon E \to \mathbb{R}$.

This geometric structure directly captures the physical reality of robotic systems. The state-dependent measurement quality is naturally represented through varying fiber dimensions and geometry across the state space. For instance, in robotic manipulation, joint angle sensing accuracy often depends on the arm configuration $q \in Q$, captured by the fiber structure variation over the configuration manifold. The connection form $\omega$ ensures consistent propagation of these measurement uncertainties while respecting the underlying physical dynamics.

The bundle structure preserves key physical properties: mechanical energy conservation through the symplectic structure, and sensor-state correlations through the horizontal distribution. This enables measurement-consistent dynamics learning via constrained neural ODEs:

$$\dot{\hat{f}} = -L_1(\hat{f} - f) + \lambda R_{fiber}(\nabla \Phi) \tag{46}$$

The measurement-aware control barrier functions (mCBFs) automatically adapt safety margins based on measurement confidence:

$$b(x, y) = b_0(x, h(x)) - L_b \|y - h(x)\| \tag{47}$$

A key strength of this geometric approach lies in its coordinate independence and preservation of physical symmetries. The intrinsic nature of the bundle structure means the framework applies equally well across different robot representations while automatically preserving important physical invariants like $SE(3)$ symmetry. This enables broad generalization across diverse robotic systems and tasks without requiring task-specific modifications to the underlying mathematical framework.

The fiber bundle structure provides robust safety guarantees through its geometric mechanisms. The bundle projection ensures safety constraints are satisfied for all possible measurements within each fiber, while the connection form enables consistent propagation of these constraints as the system evolves. This geometric approach to safety is fundamentally different from traditional methods using uncertainty ellipsoids or tubes, as it captures the intrinsic coupling between measurements and dynamics that naturally arise in physical systems.

While the framework requires sufficient smoothness of the underlying manifold $M$ and bounded measurement uncertainty, these requirements align well with numerous practical dynamics (for example, robotic systems). The geometric decomposition into horizontal and vertical components often leads to efficient computations despite the additional dimensions introduced by the bundle structure. Future extensions could enable learning optimal fiber structures from data or handling coupled uncertainties in multi-agent systems while maintaining the core geometric principles that provide rigorous safety guarantees.

This geometric perspective offers both theoretical guarantees and practical insights for implementing robust safety-critical control under uncertainty. By providing a unified treatment of measurement uncertainty that respects physical constraints and symmetries, the framework bridges the gap between theoretical safety guarantees and practical implementation. The natural handling of both physical constraints and measurement uncertainty through intrinsic geometric structures makes it particularly valuable for real-world applications where ensuring safety under imperfect sensing is crucial.

