# OpenReview forum: "Learning Dynamics under Environmental Constraints via Measurement-Induced Bundle Structures"
_ICML.cc/2025/Conference — ICML 2025 spotlightposter_

### Official Review · Reviewer_Bp16 · 2025-03-09

**Overall Recommendation:** 2

**Summary:**

This paper presents a novel geometric framework for learning unknown dynamics under environmental constraints when constraint information is only locally available and uncertain. The authors introduce a fiber bundle structure over the state space that unifies measurements, constraints, and dynamics learning. This geometric approach enables measurement-aware Control Barrier Functions (mCBFs) that adapt to local sensing conditions. By integrating Neural ODEs, the framework learns continuous-time dynamics while preserving geometric constraints, with theoretical guarantees of learning convergence and constraint satisfaction dependent on sensing quality. The authors demonstrate through simulations that their approach significantly improves both learning efficiency and constraint satisfaction compared to traditional methods, especially under limited and uncertain sensing conditions.

**Claims And Evidence:**

The claims made in the submission are supported by both theoretical analysis and experimental evidence. The authors claim that their geometric framework provides a unified approach to handling measurement uncertainty, system dynamics, and constraints, which is substantiated by the detailed mathematical formulation of the fiber bundle structure and its properties. The claim that their approach leads to improved learning efficiency and constraint satisfaction is supported by experimental results, though the paper would benefit from more detailed quantitative comparisons with baseline methods. The theoretical guarantees of learning convergence and constraint satisfaction are rigorously derived from the geometric properties of the bundle structure, providing a solid foundation for the practical implementation of the framework.

**Essential References Not Discussed:**

N/A

**Experimental Designs Or Analyses:**

The paper mentions "extensive simulations" that demonstrate significant improvements in learning efficiency and constraint satisfaction, but the details of these experiments are somewhat limited in the sections I was able to review. The experimental design appears to involve learning dynamics models under various measurement uncertainty conditions, with comparisons to traditional methods. However, without more specific information about the simulation environments, performance metrics, and statistical analyses, it is difficult to fully assess the validity of the experimental claims. The paper would benefit from more detailed descriptions of the experimental setup and comprehensive results.

**Methods And Evaluation Criteria:**

The proposed methods are mathematically sound and well-suited to the problem of learning dynamics under environmental constraints with uncertain measurements. The authors develop a comprehensive geometric framework based on fiber bundle theory, which provides a natural setting for handling measurement uncertainty. The evaluation criteria include both theoretical analysis of convergence and safety guarantees, as well as practical demonstrations of the framework's effectiveness in simulation environments. The paper would benefit from more explicit descriptions of the baseline methods used for comparison and clearer metrics for quantifying improvements in learning efficiency and constraint satisfaction.

**Other Comments Or Suggestions:**

- The paper would benefit from more illustrative examples or case studies to demonstrate the practical application of the theoretical framework.
- A more detailed discussion of the computational aspects of implementing the framework, particularly for high-dimensional systems, would strengthen the paper.
- The authors could consider providing more intuitive explanations of the key geometric concepts to make the paper more accessible to a broader audience.
- Additional visualizations of the fiber bundle structure and its relationship to measurement uncertainty would help readers better understand the geometric framework.

**Other Strengths And Weaknesses:**

Strengths:
- The paper presents a novel and mathematically rigorous geometric framework that unifies measurements, constraints, and dynamics learning.
- The theoretical guarantees of learning convergence and constraint satisfaction provide a solid foundation for the practical implementation of the framework.
- The approach naturally adapts to local sensing conditions, making it particularly suitable for real-world applications with uncertain measurements.
- The integration with Neural ODEs enables learning continuous-time dynamics while preserving geometric constraints.

Weaknesses:
- The paper is highly theoretical and may be challenging for readers without a strong background in differential geometry and control theory.
- The experimental validation could be more comprehensive, with clearer comparisons to baseline methods and more detailed quantitative results.
- The practical implementation details of the framework, particularly for complex systems, are somewhat limited.
- The paper does not extensively discuss the computational complexity of the approach or potential scalability issues.

**Questions For Authors:**

- How does the computational complexity of your approach scale with the dimensionality of the state space and the complexity of the constraints? Are there practical limitations to applying this framework to high-dimensional systems?
- The paper focuses on theoretical guarantees and simulation results. Have you considered or tested the application of this framework on physical systems with real sensor measurements? What additional challenges might arise in such settings?
- The measurement-adapted Control Barrier Functions (mCBFs) adapt to local sensing conditions. Could you elaborate on how this adaptation mechanism performs in environments with highly variable sensing quality, such as those with occlusions or sensor failures?

**Relation To Broader Scientific Literature:**

The paper effectively situates its contributions within the broader scientific literature on safety-critical control, geometric learning, and measurement-aware control. The authors provide a comprehensive review of related work in Section 2, acknowledging foundational contributions in differential geometry (Ehresmann, 1950; Kobayashi & Nomizu, 1996), control barrier functions (Ames et al., 2019), and geometric learning (Chen et al., 2018). They clearly articulate how their approach addresses limitations in existing methods, particularly in handling measurement uncertainty as an intrinsic geometric property rather than an external disturbance. The paper builds upon and extends several important lines of research in a coherent and well-motivated manner.

**Theoretical Claims:**

The paper makes several significant theoretical claims, including the formulation of measurement-adapted Control Barrier Functions, the convergence of the learning dynamics, and probabilistic safety guarantees. I have examined the theoretical development in Sections 3 and 4, including the key Theorem 3.1 which provides probabilistic safety guarantees. The proof sketch outlines a three-step approach that appears sound, though the full proof is referenced to be in Appendix A which was not available for review. The theoretical framework is well-grounded in differential geometry and control theory, with clear connections to established concepts such as fiber bundles, connections, and barrier functions.

---

> ### Author Rebuttal · Authors · 2025-03-29
>
> Thank you for your feedback.
>
> ## Experimental Details & Baselines
>
> Our Genesis physics engine uses task-specific configurations: Semi-implicit Euler (5e-4s timestep) with neo-Hookean material (μ=2kPa, λ=10kPa) for soft worm; RK4 (1e-2s timestep) with friction coefficients (0.15 static, 0.09 dynamic) for Franka; and RK4 (2e-3s timestep) with aerodynamic effects for quadrotor. Simulations employ continuous collision detection (3mm threshold) and sequential impulse solving (200 iterations). Statistical validation includes 95% CIs from 10 trials, with variation coefficients of 3.2-6.7%. We use Poisson disk obstacle sampling (20cm separation), bi-directional RRT for feasibility checks, and standardized evaluation protocols. While Table 1 provides comprehensive metrics across all baselines, the revised appendix will include detailed statistical analysis and simulation parameters to enhance reproducibility.
>
> ## Theoretical Framework
>
> While fiber bundle formalism necessitates rigorous mathematics, we'll add this intuitive analogy with a supporting diagram: fiber bundle as multi-story building where base space (state space) is the floor plan, fibers are vertical spaces representing possible measurements, safety certificates are structural supports, and connection forms are elevators maintaining coherence between horizontal (state) and vertical (measurement) dimensions. The key insight is that traditional approaches treat measurement uncertainty as an external disturbance, whereas our approach incorporates it as an intrinsic geometric property of the system structure. We will add a visualization of the fiber bundle structure in the revised version, showing how trajectories in state space are associated with measurement uncertainties in the total space.
>
> ## Computational Complexity & High-Dimensionality
>
> Our algorithm demonstrates excellent scalability with complexity divided into: fiber bundle connection calculation (O(n²), n=state dimension), mCBF evaluation (O(n+m), m=measurement dimension), and safety-constrained optimization (O(nd), d=policy parameters). The revised version will include a new section on computational complexity analysis with runtime measurements across system dimensions (n=6: 1.2ms, n=12: 3.8ms, n=24: 9.5ms, n=48: 22.7ms), plus IEEE 14-bus power grid validation tests using PYPOWER in the appendix, where our method achieved 97.2% constraint satisfaction under line thermal limits of 85% and voltage stability constraints of ±0.05 p.u., demonstrating effective application to practical higher-dimensional systems.
>
> ## Physical Validation on real Franka Arm
>
> Our revision conducted validation on a 7-DOF Franka arm in the real world with joint velocity limits (±1.0 rad/s), acceleration constraints (±2.0 rad/s²), end-effector pose stability (±5° deviation), force limits (5N), and power consumption constraints (≤60W). We applied Gaussian noise, dropouts, and delays to IMU/force sensors. Results show 93.5% constraint satisfaction versus 81.2% for baseline methods, and trajectory error of 2.3cm compared to the baseline's 4.8cm. The revised paper will include detailed experimental setup photos, hardware specifications, and constraint visualization plots. We identified challenges, including joint friction, inconsistent sensor frequencies, and constraint priority conflicts. Our geometric framework naturally addresses these via dynamic constraint better priority adjustment based on measurement quality compared with other methods.
>
> ## Measurement Quality Adaptation
>
> We created controlled testing scenarios with noise levels (0-8%) and data dropout rates (0-25%) for the simulation Franka task. Results show safety boundary adaptations: high-quality regions (noise <2%, dropout <5%) at 3.5cm, medium uncertainty (noise 4-6%) at 5.7cm, high uncertainty (noise >7%, dropout >20%) at 8.2cm, with fallback to conservative behavior during sensor failure. This geometric adaptation (from Equation 13) reduced path length by 25.3% versus fixed-boundary methods while maintaining safety, confirming the fiber bundle framework's effectiveness for varying measurement quality. The revised version will add an adaptive safety boundary analysis figure showing boundary values as a continuous function of measurement quality with corresponding robot trajectories under different sensing conditions.
>
> ## Appendix Accessibility
>
> The complete PDF contains all appendices, including the proof of Theorem 3.1 through six steps: preliminary assumptions, perfect measurement invariance, uncertainty propagation, local certification, temporal correlation analysis, and global safety guarantees. We can provide a separate file if needed.
>
> ## Conclusion
>
> We believe these clarifications and additional visualizations in the revised version will improve readability while confirming the effectiveness and practicality of our method through real system validation.

---

### Official Review · Reviewer_xLW4 · 2025-03-14

**Overall Recommendation:** 5

**Summary:**

This paper considers the problem of learning unknown dynamics models in the presence of model constraints. The paper points out that classical treatments of this problem ignore the system's inherent geometry while taking measurements into account and, in doing so, ignore important information that could be useful during learning. However, measurement uncertainty induces a fiber bundle structure which naturally lends itself to neural ODE models. Positive results are shown in an extensive empirical study.

**Claims And Evidence:**

*Claims*
1. Proposes a novel geometric framework that unifies measurement uncertainty, system dynamics, and constraints within fiber bundle structures
2. Introduces adaptive measurement aware safety certificates that adjust conservative margins based on local measurement quality.
3. Demonstrates enhanced generalization capabilities across different scenarios without requiring global information through experimental validation.
4. The proposed method is theoretically sound; learning converges with a certificate of safety.

*Evidence*
1. This is supported with the formalism in section 3. Further support comes from the context provided by related work.
2. This is also supported with the formalism in section 3.
3. Experiments with three simulated environments provide the data supporting this claim. The proposed method consistently performs well when compared with a variety of metrics and among relevant state-of-the-art baseline methods.
4. Evidence comes from Theorems 3.1 and 4.1.

**Essential References Not Discussed:**

I did not identify any essential references that were omitted.


Below are some pointers to non-essential related work---references the authors may find interesting.

*Learning sensor geometry*
1. [Map learning with uninterpreted sensors and effectors](https://www.sciencedirect.com/science/article/pii/S0004370296000513)
2. [Discovering sensor space: Constructing spatial embeddings that explain sensor correlations](https://ieeexplore.ieee.org/document/5578854)
3. [Adapting the Function Approximation Architecture in Online Reinforcement Learning](https://arxiv.org/pdf/2106.09776)

**Experimental Designs Or Analyses:**

I thoroughly reviewed the experiments, their methodology, results, and analysis. More comments can be found in my response to methods.

**Methods And Evaluation Criteria:**

Experiments evaluate the proposed method in three simulation environments.
1. Worm robot
2. Manipulator arm
3. Quadrotor drone

Uniformly positive results among these suggest an algorithm's ability to perform well across a variety of settings.

Performance is evaluated with

- Success rate
- Path efficiency measures
- Safety margins
- Control quality

Within these settings, experiments consider four kinds of evaluations.

*Learning-based Safety Certification:* Methods with fixed barrier functions achieve reasonable success but come second to the proposed method.

*Physics-informed and Geometric Methods:*  The proposed method outperforms these baselines in this environment, but the paper does not explain why. The baseline was introduced as a check of "physical consistency." What that means precisely is left unsaid. Further explanation of this result is needed to avoid the baseline being viewed as a strawman.

*Robust and Adaptive Control:* Baselines achieve high rates of constraint satisfaction while producing overly conservative trajectories. The proposed method avoids this pathology.

*Uncertainty-Aware Predictive Control:* The proposed method ranks highest under this evaluation due to its integration of relevant measurement uncertainty with constraints.

Finally, the paper performs an ablation of several aspects of their method to demonstrate the relative performance gain of each feature.


*Assessment :* Overall the experiments were thorough, comprehensive, well-documented, of sound methodology, and demonstrate consistently positive results.

**Other Comments Or Suggestions:**

I have no additional comments to add here.

**Other Strengths And Weaknesses:**

*Strengths*
- This paper checks about every box
- Originality: The paper appears meaningfully novel, both theoretically and from a practical perspective.
- Clarity: The paper is exceptionally clear. I expect that even non-experts from this area will be able to grasp its main contributions.
- Significance: Based on the theoretical results and the extensive empirical validation, I think it's fair to say this paper makes a significant contribution toward algorithms that learn how systems move and behave, while ensuring they stay safe, even when they only have limited information about their surroundings.

*Weaknesses*
- The material is quite technical and, as a result, unnecessarily restrict its potential audience. It may be possible to appeal to more readers by elaborating on key concepts, such as the three mCBF conditions.

**Questions For Authors:**

I have no questions for the authors.

**Relation To Broader Scientific Literature:**

This paper is well-positioned with respect to related work. The paper does a good job of clearly covering the broader areas of research which it intersects, and providing descriptions of more closely-related works.

**Theoretical Claims:**

*Claims*

The paper makes two main theoretical claims.

1. Theorem 3.1 (paraphrased): Whenever the system starts under safe conditions, the probability that the state remains safe is bounded below.
2. Theorem 4.1 (paraphrased): The proposed learning dynamics converge in a safe manner.

*Evidence*
1. The proof of this involved several intermediate results. Each was presented clearly and appears sound.
2. The proof of this was easy to follow and appears sound. The proof makes an assumption about real-valued eigenvalues. The conditions under which this assumption hold should qualify the theorem statement.

*Minor comment about the formalism*
- The dependence of $u$ is missing from (9).

---

> ### Author Rebuttal · Authors · 2025-03-30
>
> We appreciate your evaluation and suggestions. Your positive assessment is encouraging, and we accept the improvements you suggested. Below are the enhancements we plan to implement:
>
> ## Correction of Theoretical Statements
>
> We will add the dependency on $\alpha$ in equation (9), correcting it to $\inf_{u\in U}[L_fb + L_gbu + \alpha(b)] \geq 0$ to ensure mathematical rigor. In the statement of Theorem 4.1, we will specify the conditions for the real-valued eigenvalue assumption. We will add: "Assume that $\mathcal{L}_1$ is a symmetric positive definite operator" (so it only has positive eigenvalues) and discuss in which practical systems this assumption naturally holds, such as in mechanical systems with physical dissipation processes.
>
> ## Enhancing Accessibility of Technical Content
>
> We will make the technical content accessible by enhancing the intuitive explanation of mCBF conditions. Condition 1 ($b(x,y) \geq 0 \Rightarrow x \in S_0$) ensures that positive values of the safety certificate directly correspond to physically safe system states; Condition 2 ($\inf_{u\in U}[L_fb + L_gbu + \alpha(b)] \geq 0$) guarantees that system dynamics automatically "avoid" unsafe region boundaries, with stronger "avoidance force" as boundaries are approached; Condition 3 ($|b(x,y_1) - b(x,y_2)| \leq L_bd_Y(y_1,y_2)$) ensures that measurement noise does not cause drastic changes in safety certificates, providing robustness against measurement uncertainty. We will add a key visualization showing how trajectories in the state space M intersect with measurement uncertainty fibers in the total space E, and how the connection form transfers tangent vectors from the base space to the fibers, demonstrating the difference between our method and traditional approaches in handling measurement uncertainty.
>
> ## Key Insights on Fiber Bundle Structure for Measurement Uncertainty
>
> Traditional methods and our approach have differences: traditional methods treat measurement uncertainty as external disturbances, requiring first estimating states then controlling (introducing compound errors). In contrast, our method treats measurement uncertainty as an intrinsic geometric property of the system structure, directly mapping from measurement space to control actions through the fiber bundle structure. We will add visualization of the fiber bundle structure, showing how state trajectories in the base space M generate "measurement uncertainty tubes" in the total space E, and how the connection operator $K(x)(y-h(x))$ automatically adjusts safety boundaries. We will explain how the terms $K(x)(y-h(x))$ and $\alpha\|y-h(x)\|^2\nabla_y\Phi$ in equation (12) act as natural "regulators" of measurement quality, automatically increasing conservatism in uncertain regions. These terms transform the geometric structure of measurement uncertainty into adaptive safety margins, enabling our method to enhance safety guarantees as measurement uncertainty increases.
>
> ## Improving Fairness and Relevance of Experimental Analysis
>
> Regarding the comparison with physical consistency baseline methods, we will articulate the fairness of comparisons. The PNDS method focuses on maintaining physical consistency but lacks explicit handling of measurement uncertainty; GeoPath and GEM excel at maintaining geometric invariance, but they were not designed to handle uncertainty induced by measurements. We will add detailed quantitative comparative analysis, including performance of different methods under various noise levels, changes in physical constraint violation rates as measurement noise increases, and the relationship between dynamic learning errors and measurement uncertainty. We will explain how we ensure all methods use the same neural network architecture, computational budget, initial and target state distributions, noise distributions, and system parameter uncertainties, providing a fairer basis for comparison.
>
> ## Integration of Recommended References
>
> We will create a new subsection discussing connections with sensor geometry learning research. Pierce & Kuipers' work on constructing spatial embeddings from sensor correlations provides interesting parallels to our fiber bundle structure, though our focus extends to safety guarantees under uncertainty. Their work on learning hierarchical models from uninterpreted sensorimotor signals also resonates with our approach to learning dynamics under environmental constraints. We will explore how these perspectives complement our measurement-adaptive learning methods and discuss how our framework might benefit from their insights on sensor space embedding and abstraction of continuous environments to more manageable representations.
>
> We thank you for the valuable suggestions, which help improve our work and provide guidance for our research direction. We believe that with these improvements, the paper will be clearer, more rigorous, and more accessible while maintaining its technical depth and theoretical contributions.

---

### Official Review · Reviewer_bzHQ · 2025-03-14

**Overall Recommendation:** 3

**Summary:**

This study proposes a geometric approach to learning dynamics with safety guarantees, leveraging the bundle structure to account for uncertainties. After presenting the geometric approach for controlled dynamical systems, the study introduces measurement-adapted control barrier functions, which enable the development of safety guarantees. Furthermore, the paper presents a learning framework for policy design with safety guarantees based on the bundle framework.

**Claims And Evidence:**

The safety guarantee is given in Theorem 3.1. The numerical experiments show superior performance compared to existing methods.

**Essential References Not Discussed:**

None

**Experimental Designs Or Analyses:**

There are no issues with the experimental design or analyses.

**Methods And Evaluation Criteria:**

The proposed method is based on a mathematical geometric framework, specifically leveraging bundle structures. The theoretical foundation is sound. I do not see any issues with the evaluation criteria for the numerical experiments.

**Other Comments Or Suggestions:**

In Section 4.4, the cost $J(\Theta)$ is given in a discrete-time setting. However, the system in (1) is formulated as a continuous-time system. Please clarify this inconsistency.

**Other Strengths And Weaknesses:**

This paper has its strength in proposing a geometric approach based on the bundle structure. This is interesting, but it may be necessary to improve the presentation to ensure that readers clearly understand the contribution. Specifically, it is somewhat unclear whether all the content in Section 3, such as the modeling in 3.1 and the fiber bundle framework in 3.2, is proposed in this study or has already been presented in existing papers.

**Questions For Authors:**

1. The problem setting is somewhat unclear. Given the system in (1), which includes the measurement equation, a typical setting would be a partially observed control setting. However, the policy in Section 4.4 is given in the form of $\pi_{\Theta}: \mathcal{M} \to \mathcal{U}$, which has access to the value of $x$. It is unclear why the measurement y is considered when obtaining the safety policy.
2. I was wondering if it is possible to provide a concrete example of $\mathcal{L}_1$ and $\mathcal{L}_2$ in (12). I could not fully follow the discussion in Section 4.1.

**Relation To Broader Scientific Literature:**

This study is motivated by the geometric approaches in analytical mechanics and control theory. The bundle plays an important role in these fields. In this sense, a key contribution of this study—introducing the bundle-based approach—is its role in bridging the existing results in control theory and learning.

**Theoretical Claims:**

As far as I have checked, the proofs in the appendix are sound. However, the statement of Theorem 3.1 may need to be revised. Since the control u is considered in the controlled dynamical system, the theorem’s statement should be like as follows:

Theorem 3.1. Given an $m C B F$ b satisfying the preceding conditions, if $b(x(0), y(0)) \geq 0$, then for any admissible noise sequences $w(\cdot), v(\cdot)$, **there exists $u_t$ ($t \in [0, \infty)$)**:
$$
\mathbb{P}\left(x(t) \in \mathcal{S}_0 \text { for all } t \geq 0\right) \geq 1-\exp \left(-c / \delta_v^2\right)
$$
where $c>0$ is a constant depending on system parameters.

The solution $x(t)$ depends on the control $u$. Therefore, the current form of the statement of Theorem 3.1 does not clearly specify which control $u$ is used to obtain the probabilistic bound.

---

> ### Author Rebuttal · Authors · 2025-03-30
>
> We thank you for your thoughtful feedback and have prepared appropriate revisions.
>
> ## On Theorem 3.1 Formulation
>
> We appreciate that you correctly noted that the theorem should explicitly specify the control law. We will revise Theorem 3.1 to clearly indicate that there exists a control strategy $\pi(x,y) \in \Pi_{\text{safe}}$ for which the safety probability bound holds. Specifically, $\Pi_{\text{safe}} =$ \{$\pi : M \times Y \to U \mid L_e b(x,y,\pi(x,y)) + \alpha(b(x,y)) \geq 0, \forall(x,y) \in E$\}, where $L_e b$ is the extended Lie derivative along the fiber bundle connection, defined as $L_e b(x,y,u) = \nabla_x b(x,y)^T f(x,u) + \nabla_y b(x,y)^T \frac{\partial h}{\partial x}f(x,u)$ in Section 3.6. This clarifies that safety guarantees apply to any control strategy respecting the mCBF condition.
>
> ## Clarification of Contributions in Section 3
>
> The system model in Section 3.1 builds on standard stochastic control formulations, but our innovation lies in integrating it with the fiber bundle perspective specifically for measurement uncertainty. Section 3.2's fiber bundle framework represents our original contribution, particularly the connection form in Equation (3) that couples measurement uncertainty with state dynamics through $K(x)$. To our knowledge, applying fiber bundles to measurement uncertainty in safety-critical learning is novel.
>
> ## Time Setting Consistency Issue
>
> Regarding the apparent inconsistency between continuous-time dynamics and discrete-time cost, we can formalize this transition as follows: Define continuous-time cost $J_c(\Theta) = \int_0^\infty e^{-\rho t} c(x(t),\Theta(x(t)))dt$ and discrete-time cost $J_d(\Theta) = \sum_{k=0}^\infty \gamma^k c(x(k\Delta t),\Theta(x(k\Delta t)))$, where $\gamma = e^{-\rho \Delta t}$. For small $\Delta t$, we have $|J_c(\Theta) - J_d(\Theta)| \leq C \cdot \Delta t$. Similarly, if continuous-time systems satisfy the mCBF condition $L_e b(x,y,\pi(x,y)) + \alpha(b(x,y)) \geq 0$, discrete systems maintain safety under $(b(x_{k+1},y_{k+1}) - b(x_k,y_k))/\Delta t + \alpha(b(x_k,y_k)) \geq \beta \cdot \Delta t$, where $\beta$ depends on second-order derivatives of system dynamics.
>
> ## Clarification of Problem Setting
>
> Traditional methods use the separation principle: estimate state $\hat{x} = E[x|y_{1:t}]$ then apply controller $u = \pi(\hat{x})$. Under high uncertainty, estimation error $e = x - \hat{x}$ may violate safety constraints. Our approach constructs a safety framework directly on fiber bundle $E = M \times Y$ with control strategy $\pi : M \times Y \to U$, incorporating measurement uncertainty into the safety condition: $\nabla_x b(x,y) \cdot f(x,\pi(x,y)) + \nabla_y b(x,y) \cdot \dot{h}(x) + \alpha(b(x,y)) \geq 0$. This avoids intermediate state estimation and automatically adjusts safety margins with measurement quality—as $\|y - h(x)\|$ increases, the control becomes more conservative.
>
> ## Concrete Examples of Learning Operators
>
> For dynamics learning operator $L_1$ in (12), we implement it on a 2D system where $x = [p, \xi]^T$, $f(x,u) = [\xi, u]^T$, and observation $y = h(x) + w$ with $y \in \mathbb{R}^m$. Here, $h: \mathbb{R}^2 \rightarrow \mathbb{R}^m$ is the observation function. The bundle structure comes from the relationship between states and measurements:
>
> $
> L_1(f_{\theta} - f)(x,y) = \nabla_x(f_{\theta} - f) \cdot f(x,u) + K(x)(y - h(x))
> $
>
> where $K(x) \in \mathbb{R}^{2 \times m}$ is the gain matrix. This operator works along two directions: horizontally through state dynamics (first term) and vertically through measurement fibers (second term). In implementation, the parameter update is:
>
> $\frac{d\theta}{dt} = -\Gamma \sum_{i} [(f_{\theta}(x_i) - \dot{x}i)^T \Sigma_i^{-1} \frac{\partial f_{\theta}}{\partial \theta} + \lambda K(x_i)(y_i - h(x_i))^T \frac{\partial f_{\theta}}{\partial \theta}]$
>
> Similarly, safety certificate operator $L_2$ in (12) for constraint $h_s(x) = x_{safe} - p \geq 0$ becomes:
>
> $L_2(\Phi - \Phi^*)(x,y) = \nabla_x (\Phi - \Phi^*) \cdot f(x,u) + \alpha ‖y - h(x)‖^2 \nabla_y(\Phi - \Phi^*)$
>
> where $h_s: \mathbb{R}^2 \rightarrow \mathbb{R}$ is a scalar safety constraint on position. This approach automatically adapts safety margins with measurement uncertainty. Consider position control with noisy sensors: traditional methods first estimate position and then control (introducing compounding errors), while our operators $L_1, L_2$ directly map measurements to controls, becoming more conservative as uncertainty increases through terms like $\alpha‖y-h(x)‖^2$. Our 2D navigation system benchmark in the revision confirms this advantage: with noise $\sigma_y=0.1$, dynamics learning error improves by 47% (0.08 vs 0.15); at $\sigma_y=0.5$, improvement reaches 60% (0.17 vs 0.42). For safety near obstacle at $x_{obs}=2$ with $\sigma_y=0.5$, our method maintains a 0.47 margin while standard CBFs drop to 0.22 with violations - showing how Equation 12 handles measurement uncertainty without separate estimation steps.

---

### Decision · Program_Chairs · 2025-05-01

**Decision:**

Accept (spotlight poster)

**Comment:**

This paper takes a geometric perspective on learning and control for systems with safety constraints. The reviewers appreciate the strength of this contribution, which is theoretically solid while also achieving impressive performance on numerical experiments. There is consensus among the two reviewers who engaged in discussions, while the third review is more surface level (and was flagged as likely AI generated).